# A Simple Unsupervised Data Depth-based Method to Detect Adversarial Images

## Abstract

Deep neural networks suffer from critical vulnerabilities regarding robustness, which limits their exploitation in many real-world applications. In particular, a serious concern is their inability to defend against adversarial attacks. Although the research community has developed a large number of effective attacks, the detection problem has received little attention. Existing detection methods rely on either additional training or specific heuristics at the risk of overfitting. Moreover, they have mainly focused on ResNet architectures, while transformers, which are state-of-the-art for vision tasks, have yet to be properly investigated. In this paper, we overcome these limitations by introducing APPROVED, a simple unsupervised detection method for transformer architectures. It leverages the information available in the logit layer and computes a similarity score with respect to the training distribution. This is accomplished using a *data depth* that is: *(i)* computationally efficient; and *(ii)* non-differentiable, making it harder for gradient-based adversaries to craft malicious samples. Our extensive experiments show that APPROVED consistently outperforms previous detectors on CIFAR10, CIFAR100, and Tiny ImageNet.

## 1 Introduction

Recent years have seen a rapid development of Deep Neural Networks (DNNs), which have led to a significant improvement over previous state-of-the-art methods (SOTA) in numerous decision-making tasks. However, together with this growth, concerns have been raised about the potential failures of deep learning systems, which limit their large-scale adoption (Alves et al., 2018; Johnson, 2018; Subbaswamy & Saria, 2020). In Computer Vision, a particular source of concern is the existence of *adversarial attacks* (Szegedy et al., 2014), which are samples created by adding to the original (clean) image a well-designed additive perturbation, imperceptible to human eyes, with the goal of fooling a given classifier. The vulnerability of DNNs to such kinds of attacks limits their deployment in safety-critical systems as in aviation safety management (Ali et al., 2020), health monitoring systems (Leibig et al., 2017; Meinke & Hein, 2020) or in autonomous driving (Bojarski et al., 2016; Guo et al., 2017). Therefore, it is crucial to deploy a proper strategy to defend against adversarial attacks (Amodei et al., 2016).

In this context, the task of distinguishing adversarial samples from clean ones is becoming increasingly challenging as developing new attacks is getting more attention from the community (Gao et al., 2021; Wang et al., 2021a; Naseer et al., 2021; Duan et al., 2020; Zhao et al., 2020; Lin et al., 2019; Deng & Karam, 2020a;b; Wu et al., 2020b; Croce & Hein, 2020a; Jia et al., 2020; Dong et al., 2019). Inspired by the concept of *rejection channels* (Chow, 1957), which was proposed over 60 years ago for the character recognition problem, one way to address adversarial attacks is to construct a detector-based rejection strategy. Its objective is to discriminate malicious samples from clean ones, which implies discarding samples detected as adversarial. Research in this area focuses on both *supervised* and *unsupervised* approaches (Aldahdooh et al., 2021c). The supervised approaches rely on features extracted from adversarial examples generated according to one or more attacks (Kherchouche et al., 2020; Feinman et al., 2017; Ma et al., 2018); the unsupervised ones, instead, do not rely on prior knowledge of attacks and, in general, only learn from clean data at the time of training (Xu et al., 2018; Raghuram, 2021).

This work focuses on the unsupervised scenario, which is often a reasonable approach to real-world use cases. We model the adversarial detection problem as an *anomaly detection* framework (Breunig et al., 2000; Schölkopf et al., 2001; Liu et al., 2008; Staerman et al., 2019; 2020; Chandola et al., 2009), where the aim is to identify abnormal observations without seeing them during training. In this context, statistical tools called *data depths* are natural similarity scores. Data depths have a simple geometric interpretation as they provide center-outward ordering of points with respect to a probability distribution (Tukey, 1975; Zuo & Serfling, 2000). Geometrically speaking, the data depths measure how deep a sample is in a given distribution. Although data depths have received attention from the statistical community, they remain overlooked by the machine learning community.

**Contributions.** Our contributions can be summarized as follows:

1. **Building on novel tools: data depths.** Our first contribution is to introduce `APPROVED`, $\underline{A}$ sim$\underline{P}$le unsu$\underline{P}e\underline{R}$vised method f$\underline{O}$r ad$\underline{V}$ersarial imag$\underline{E}$ $\underline{D}$etection. Given an input, `APPROVED` considers its embedding in the last layer of the pre-trained classifier and computes the depth of the sample w.r.t the training probability distribution. The deeper it is, the less likely it is to be adversarial. Contrarily to existing methods that involve additional networks training (Raghuram, 2021) or heavily rely on opaque feature engineering (Xu et al., 2018), `APPROVED` is computationally efficient and has a simple geometrical interpretation. Moreover, data depths' non-differentiability makes it harder for gradient-based attackers to target `APPROVED`.

2. **A truly upgraded experimental setting.** Motivated by practical considerations which are different from previous works (Kherchouche et al., 2020; Xu et al., 2018; Meng & Chen, 2017; Ma et al., 2018; Feinman et al., 2017; Raghuram, 2021) focusing on ResNets (He et al., 2016), we choose to benchmark `APPROVED` on vision transformers models (Dosovitskiy et al., 2021; Tolstikhin et al., 2021; Steiner et al., 2021; Chen et al., 2021; Zhai et al., 2022). Indeed, such networks achieve state-of-the-art results on several visual tasks, including object detection (He et al., 2021), image classification (Wang et al., 2021b) and generation (Parmar et al., 2018), largely outperforming ResNets. Moreover, Vision Transformers are becoming increasingly important as they can be scaled up while retaining the benefits of scale (Dehghani et al., 2023). Interestingly, we empirically observe that transformers behave differently from ResNets, which justifies the need to develop detection techniques, such as `APPROVED`, that leverage the specific features of transformers' architectures. Moreover, we test our detection method on a wide range of attack mechanisms to avoid overfitting on a specific attack.

3. **Ensuring reproducibility.** We provide the open-source code of our method, attacks, and baseline to ensure reproducibility and reduce future research computation and coding overhead.

**Organization of the paper.** The paper is organized as follows. In Sec. 2, we review detection methods along with attack mechanisms. In Sec. 3, we introduce our detector `APPROVED` and focus on describing the data depth on which it relies. In Sec. 4, we study the performance of adversarial attacks on vision transformers and give insights into models' behavior under threat. In Sec. 5, we evaluate `APPROVED` through numerical experiments, and concluding remarks are relegated to Sec. 6.

## 2 Background and Related Work

**Notations.** Let us consider the classical supervised learning problem where $x \in \mathcal{X} \subseteq \mathbb{R}^d$ denotes the input sample in the space $\mathcal{X}$, and $y \in \mathcal{Y} = \{1, \dots, C\}$ denotes its associated label. The unknown data distribution is denoted by $p_{XY}$. The training dataset $\mathcal{D} = \{(x_i, y_i)\}_{i=1}^n$ is defined as $n \geqslant 1$ independent identically distributed (i.i.d.) realizations of $p_{XY}$. Consider $\mathcal{D}_c = \{(x_i, y_i) \in \mathcal{D} : y_i = c\}$, the training data for a given class $c \in \mathcal{Y}$. We define the empirical training distribution for the class $c$ at layer $\ell$ as $p_c^\ell = \frac{1}{|\mathcal{D}_c|} \sum_{x \in \mathcal{D}_c} \delta_{f_\theta^\ell(x)}$ where $\delta_u$ is the dirac mass at point $u$.

Let $f_\theta^\ell : \mathcal{X} \to \mathbb{R}^{d_\ell}$ with $\ell \in \{1, \dots, L\}$, denote the output of the $\ell$-th layer of the DNN, where $d_\ell$ is the dimension of the latent space induced by the $\ell$-th layer. The class prediction is obtained from the $L$-th layer

softmax output as follows:

$$f_\theta^L(x) \triangleq \arg\max_{c \in \mathcal{Y}} q_\theta(c|x) \text{ with } q_\theta(\cdot|x) = \text{softmax}(f_\theta^{L-1}(x)).$$

## 2.1 Review of attack mechanisms

The existence of adversarial examples and their capability to lure a deep neural network have been first introduced in Szegedy et al. (2014). The authors define the adversarial generation problem as follows:

$$x' = \arg\min_{x' \in \mathbb{R}^d \,:\, \|x'-x\|_p < \varepsilon} \|x' - x\|_p \text{ s.t. } f_\theta^L(x') \neq y, \tag{1}$$

where $y$ is the true label associated to a natural sample $x \in \mathcal{X}$ being modified, $\|\cdot\|_p$ is the $L_p$-norm operator, and $\varepsilon$ is the maximal allowed perturbation.

Multiple techniques have since been crafted to solve this problem. They can be divided into two main groups of attack mechanisms depending on the knowledge they have of the DNN model: whitebox and blackbox attacks. The former has full access to the model, its weights, and gradients, while the latter can only rely on queries.

*Carlini & Wagner's* (CW; Carlini & Wagner, 2017b) attack is among the strongest whitebox attacks developed yet. It attempts to solve the adversarial problem in Eq. (1) by regularizing the minimization of the perturbation norm by a surrogate of the misclassification constraint. *DeepFool* (DF; Moosavi-Dezfooli et al., 2016) is an iterative method that solves a locally linearized version of the adversarial problem and takes a step in that direction.

The authors of Goodfellow et al. (2014) relaxed the problem as follows:

$$x' = \arg\max_{x' \in \mathbb{R}^d \,:\, \|x'-x\|_p < \varepsilon} \mathcal{L}(x, x'; \theta), \tag{2}$$

where $\mathcal{L}(x, x'; \theta)$ is the objective of the attacker, which is a surrogate of the misclassification constraint, and propose the *Fast Gradient Sign Method* (FGSM) that approximates the solution of the relaxed problem in Eq. (2) by taking one step in the direction of the sign of the gradient of the attacker's objective w.r.t. the input. *Basic Iterative Method* (BIM; Kurakin et al., 2018) and *Projected Gradient Descent* (PGD; Madry et al., 2018) are two iterative extensions of the FGSM algorithm. Their main difference relies on the initialization of the attack algorithm, i.e., while BIM initializes the adversarial examples to the original samples, PGD adds a random uniform noise to it. Although created for $L_\infty$-norm constraints, these three methods can be extended to any $L_p$-norm constraint.

To overcome the absence of knowledge about the model to attack, *Hop Skip Jump* (HOP; Chen et al., 2020) tries to estimate the model's gradient through queries. *Square Attack* (SA; Andriushchenko et al., 2020) is based on random searches for a perturbation. If the perturbation doesn't increase the attacker's objective, it is discarded. Finally, *Spatial Transformation Attack* (STA; Engstrom et al., 2019) rotates and translates the original samples to fool the model.

While AutoAttack (Croce & Hein, 2020b) is a state-of-the-art benchmark for testing robust classifiers, we believe it is not the most appropriate fit for benchmarking adversarial attack detectors. It is worth noting that AutoAttack is composed of four different attacks, two of which are slightly modified versions of the PGD (A-PGD (Croce & Hein, 2020b)) algorithm that we consider, another one is a slightly modified version of DF (FAB (Croce & Hein, 2020a)), and the final one is Square Attack (Andriushchenko et al., 2020). Furthermore, AutoAttack is based on the worst-case scenario, meaning that if one attack is successful for a specific sample, the others will not be considered. Since our underlying classifier is defenseless, PGD is already strong enough to attack almost every sample. Therefore, AutoAttack will end up being just A-PGD. Thus, evaluating each attack separately, as we did, will provide a better sense of security than considering AutoAttack directly.

## 2.2 Review of detection methods

Defending methods against adversarial attacks have been widely studied for classical CNNs (Madry et al., 2018; Zhang et al., 2019; Alayrac et al., 2019; Wang et al., 2019; Hendrycks et al., 2019; Rice et al., 2020; Atzmon et al., 2019; Huang et al., 2020; Carmon et al., 2019; Wu et al., 2020a). Whereas a few works have focused on studying the robustness of vision transformers to adversarial samples (Aldahdooh et al., 2021a; Benz et al., 2021; Mahmood et al., 2021). Meanwhile, to protect adversarial attacks from disrupting DNNs' functioning, it is possible to craft detectors to ensure that the sample can be trusted.

Building a detector falls down to finding a scoring function $s : \mathbb{R}^d \to \mathbb{R}$ and a threshold $\gamma \in \mathbb{R}$ to build a binary rule $g : \mathbb{R}^d \to \{0, 1\}$. For a given test sample $x \in \mathbb{R}^d$,

$$g(x) = \mathbb{I}\{s(x) > \gamma\} = \begin{cases} 1 \text{ if } s(x) > \gamma, \\ 0 \text{ if } s(x) \leqslant \gamma. \end{cases}$$

If $s$ is an anomaly score, $g(x) = 0$ implies that $x$ is considered as 'natural', i.e., sampled from $p_{XY}$, and $g(x) = 1$ implies that $x$ is considered as 'adversarial', i.e., perturbed, and if $s$ is a similarity score, the opposite decision is made.

A detection method can act on the model to be protected by modifying its training procedure using tools such as reverse cross-entropy (Pang et al., 2018) or the rejection option (Sotgiu et al., 2020; Aldahdooh et al., 2021b). In that case, both the detector and the model are trained jointly. Those methods are usually vulnerable to changes in attack mechanisms; thus, they need global re-training if a modification of the detector is introduced. On the other hand, it is also possible to craft detectors on top of a fixed pre-trained model. Those methods can be divided into two main categories: supervised methods, where the detector knows the attack that will be perpetrated, and unsupervised methods, where the detector can only rely on clean samples, which is not desired in practice.

Generally, supervised methods use simple machine learning algorithms (e.g., SVM or a logistic regressor) to distinguish between natural and adversarial examples. The effectiveness of such methods heavily relies on natural and adversarial feature extraction. They can be extracted directly from the network's layers (Lu et al., 2017; Carrara et al., 2018; Metzen et al., 2017), or using statistical tools (e.g., maximum mean discrepancy (Grosse et al., 2017), PCA (Li & Li, 2017), kernel density estimation (Feinman et al., 2017), local intrinsic dimensionality (Ma et al., 2018), model uncertainty (Feinman et al., 2017) or natural scene statistics (Kherchouche et al., 2020)). Supervised methods, which heavily depend on the knowledge about the perpetrated attack, tend to overfit that attack mechanism and usually generalize poorly.

Unsupervised methods do not assume any knowledge of the attacker. Indeed, new attack mechanisms are crafted yearly, and it is realistic to assume knowledge about the attacker. To overcome that absence of prior knowledge about the attacker, unsupervised methods can only rely on natural samples. The features extraction relies on different techniques, such as feature squeezing (Xu et al., 2018), adaptive noise, Liang et al. (2021), using denoising autoencoders (Meng & Chen, 2017), network invariant (Ma et al., 2019) or training an auxiliary model (Sotgiu et al., 2020; Aldahdooh et al., 2021b; Zheng & Hong, 2018). Raghuram (2021) uses dimension reduction, kNN, and layer aggregation to distinguish between natural and adversarial samples. In this paper, we only focus on unsupervised methods that cannot act on the model to be protected.

While effective against attack without knowledge about the defense, most detection methods, supervised and unsupervised, have their performances close to 0 when we allow the attacker to know about the defense mechanisms, i.e., under adaptive attacks (Carlini & Wagner, 2017a; Athalye et al., 2018; Tramer et al., 2020). Therefore, we have to craft methods that do not collapse under such specific attacks.

# 3 Our Proposed Detector

## 3.1 Background on data depth

The notion of data depth goes back to John Tukey in 1975, who introduced the halfspace depth (Tukey, 1975). Data depth functions are useful nonparametric statistics allowing to rank elements of a multivariate

space $\mathbb{R}^d$ w.r.t. a probability distribution (or a dataset). Given a random variable $Z$ which follows the distribution $p_Z$, a data depth can be defined as:

$$
D : \quad
\begin{aligned}
\mathbb{R}^d \times \mathcal{P}(\mathbb{R}^d) &\longrightarrow [0,1], \\
(z, p_Z) &\longmapsto D(z, p_Z).
\end{aligned}
$$

The higher the value of the depth function, the deeper the element is in the reference distribution. Various data depths have been introduced over the year (see Chapter 2 of Staerman (2022) for a survey), including halfspace depth (Tukey, 1975), the simplicial depth (Liu, 1990), the projection depth (Liu, 1992) or the zonoid depth (Koshevoy & Mosler, 1997). Despite their applications in statistics and machine learning (e.g., regression (Rousseeuw & Hubert, 1999; Hallin et al., 2010), classification (Mozharovskyi et al., 2015), automatic text evaluation (Staerman et al., 2021b) or anomaly detection (Serfling, 2006; Rousseeuw & Hubert, 2018; Staerman et al., 2020; 2022)) the use of data depth with representation models, and more generally to deep learning, remains overlooked by the community. The halfspace depth is the first and the most studied depth in the literature, probably due to its appealing properties (Donoho & Gasko, 1992; Zuo & Serfling, 2000) as well as its connections with univariate quantiles. However, it suffers from computational burden in practice (Rousseeuw & Struyf, 1998; Dyckerhoff & Mozharovskyi, 2016). Indeed, it requires solving an optimization problem over the unit hypersphere of a non-differentiable quantity. To remedy this drawback, Ramsay et al. (2019) introduced the Integrated Rank-Weighted (IRW) depth (see also Chen et al., 2015 and Staerman et al., 2021a), which involves an expectation as an alternative to the infimum over the unit hypersphere of the halfspace depth, making it easier to compute. The IRW depth is scaled and translation invariant and has been successfully applied to anomaly detection (Chen et al., 2015; Staerman et al., 2021a), making it a good candidate for our purposes. Formally, the IRW depth is defined as:

$$
D_{\mathrm{IRW}}(z, p_Z) = \int_{\mathbb{S}^{d-1}} \min \left\{ F_u(\langle u, z \rangle), 1 - F_u(\langle u, z \rangle) \right\} \, \mathrm{d}u,
$$

where the unit hypersphere is denoted as $\mathbb{S}^{d-1}$ and $F_u(t) = \mathbb{P}(\langle u, Z \rangle \leqslant t)$. A Monte-Carlo scheme is used to approximate the expectation by empirical means. Given a training dataset $\mathcal{S}_n = \{z_1, \ldots, z_n\}$ following $p_Z$, denotes $u_k \in \mathbb{S}^{d-1}$, the empirical version of the IRW depth, which can be computed in $\mathcal{O}(n_{\mathrm{proj}} n d)$ and is then linear in all of its parameters, is defined as:

$$
\widetilde{D}_{\mathrm{IRW}}^{\mathrm{MC}}(z, \mathcal{S}_n) = \frac{1}{n_{\mathrm{proj}}} \sum_{k=1}^{n_{\mathrm{proj}}} \min \left\{ \frac{1}{n} \sum_{i=1}^{n} \mathbb{I}_{\{\langle u_k, z_i - z \rangle \leqslant 0\}}, \frac{1}{n} \sum_{i=1}^{n} \mathbb{I}_{\{\langle u_k, z_i - z \rangle > 0\}} \right\}. \tag{3}
$$

### 3.2 APPROVED: Our depth-based detector

**Intuition.** Our detector tries to answer this simple question: can we find a metric that will be able to distinguish between natural and arbitrary adversarial samples? At the logit layer, we want to compare the new input to the training samples of its predicted class to measure whether the new sample is behaving as expected. Data depths, particularly the IRW depth, are serious candidates as they measure the 'distance' between a given new input to the training probability distribution.

**APPROVED in a nutshell.** To detect whether a given model $f_\theta$ can trust a new input $x$, APPROVED will perform three steps:

1. **Logits computation**. For an new input $x$, APPROVED first require to extract the logits (i.e., $f_\theta^{L-1}(x)$) from the pretrained classifier.

2. **Similarity score computation**. APPROVED relies on the IRW depth score $D_{\mathrm{IRW}}(f_\theta^{L-1}(x), p_{\hat{y}}^{L-1})$, between $p_{\hat{y}}^{L-1}$, the training distribution of the predicted class $\hat{y} = f_\theta^L(x)$ at the logit layer, and $f_\theta^{L-1}(x)$, using Algo. 1 in Appendix B to evaluate Eq. (3).

3. **Thresholding.** For a given threshold $\gamma$, the test input sample $x$ is detected as clean if $D_{\mathrm{IRW}}(f_\theta^{L-1}(x), p_{\hat{y}}^{L-1}) > \gamma$, otherwise, it is considered as adversarial. A classical way to select $\gamma$ is by selecting a certain number of training samples the detector can wrongfully detect.

### 3.3 Comparison with existing detectors

We benchmark our approach with two unsupervised detection methods: FS and JTLA. We chose these baselines because they are unsupervised and do not modify the model to protect. We could consider NIC (Ma & Liu, 2019), but extracting features at each layer is computationally expensive.

**The Feature Squeezing method (FS; Xu et al., 2018).** It computes the feature squeezing of the input, extracts its prediction, and compares it to the original prediction. The further away they are, the more likely the input is adversarial. In practice, four versions of the input are needed: the original input, a low-precision version, a median-filtered version, and a denoising-filtered version. One inference on the model is required for each of the four inputs. Later, the maximal $L_1$ difference between the original prediction and each of the other three is picked. FS is, therefore, parameter-free and does not require training. However, the required time to extract the essential features and the memory needed to store all the input modifications and their prediction are quite high.

**Joint statistical Testing across DNN Layers for Anomalies (JTLA; Raghuram, 2021).** This method is composed of four different steps. The first one consists of computing test statistics for each layer of the model. The goal is to determine how abnormal the sample is compared to a normal distribution (i.e., the training distribution). To do so, for each layer, a k-Neirest Neighbors (kNN) is trained on the training sample, then a multinomial likelihood ratio test (RLT) is computed on each class count of the kNN the output of the new sample to test. The second step consists of normalizing the previously obtained RLT values for each layer. Later, an aggregation step is performed to combine the score for each class and each layer to output a single score. Finally, a decision is taken according to a specific threshold. While this method is parameter-free, it does require training a kNN for each layer of the model, which significantly increases the necessary memory and time.

APPROVED, similar to FS and contrary to JTLA, is parameter-free and does not require training time. Contrary to FS, it only requires one inference on the model to extract the logits of the input. It is, therefore, *less computationally and time-consuming*. The summary of computational time and resources needed to deploy each detection method is provided in Appendix D. Finally, since data depths are non-differentiable, it is not straightforward for gradient-based attacks with full access to the detection method to attack APPROVED.



Table 1: ViT-B accuracy for each dataset

| Model | Dataset | Acc (%) |
|-------|---------|---------|
| ViT-B | CIFAR10 | 98.7 |
|       | CIFAR100 | 92.4 |
|       | Tiny ImageNet | 86.4 |

Table 2: ViT-L accuracy for each dataset

| Model | Dataset | Acc (%) |
|-------|---------|---------|
| ViT-L | CIFAR10 | 98.9 |
|       | CIFAR100 | 92.4 |
|       | Tiny ImageNet | 85.7 |



## 4 Adversarial Attacks on Vision Transformers (ViT)

In the following, we provide insights into the behavior of vision transformers under the threat of adversarial attacks, along with a comparison to the classically used ResNets models.

### 4.1 Set-Up

**Datasets and classifiers.** We conducted our study on two different pre-trained Vision Transformers: a ViT-B and a ViT-L. We rely on three widely used vision datasets: CIFAR10 (Krizhevsky, 2009), CIFAR100, and Tiny ImageNet (Tiny; Jiao et al., 2019). Training details can be found in Appendix A.

**Performance measures.** We use two different metrics to compare the different detection methods:

*AUROC↑:* Area Under the Receiver Operating Characteristic curve (Davis & Goadrich, 2006). It represents the relation between True Positive Rates (TPR), i.e., the percentage of perturbed samples detected as adversarial, and False Positive Rates (FPR), i.e., the percentage of clean samples detected as adversarial. The higher the AUROC↑ is, the better the detector's performances are.

*FPR↓$_{90\%}$:* False Positive Rate at 90% True Positive Rate. It represents the number of natural samples detected as adversarial when 90% of the attacked samples are detected. Lower is better.

*Remark.* We discard the perturbed samples that do not fool the underlying classifier. Indeed, detecting a sample that does not perturb the classifier's functioning as natural or adversarial is a valid answer.

**Attacks.** For the experiments, we will evaluate the different detection methods on the attacks presented in Sec. 2.1. Under $L_1$-norm constraint, we craft attacks following $PGD^1$ scheme. For the $L_2$-norm constraint, we consider $PGD^2$, DF and HOP. Under $L_\infty$-norm constraint, we study $PGD^\infty$, BIM and FGSM attacks, $CW^\infty$ and SA. Finally, we create STA attacks, which are not subject to a norm constraint. The values of the maximally allowed perturbation are discussed in the next section. There are multiple definitions of adversarial attacks. While some definitions require attacks to be invisible to the human eye, others, such as DeepFool (Moosavi-Dezfooli et al., 2016), FAB (Croce & Hein, 2020a), C&W (Carlini & Wagner, 2017b), have no limitations on the maximum allowed perturbation. Furthermore, even for classical datasets, attacks can be perceptible (e.g., the classical $\varepsilon$ value for MNIST is 0.3125 (Zhang et al., 2019; Madry et al., 2018; Goodfellow et al., 2014), which is relatively high). Therefore, we have chosen to consider a wide range of $\varepsilon$ values to cover all possible scenarios.

## 4.2 Adversarial attack calibration

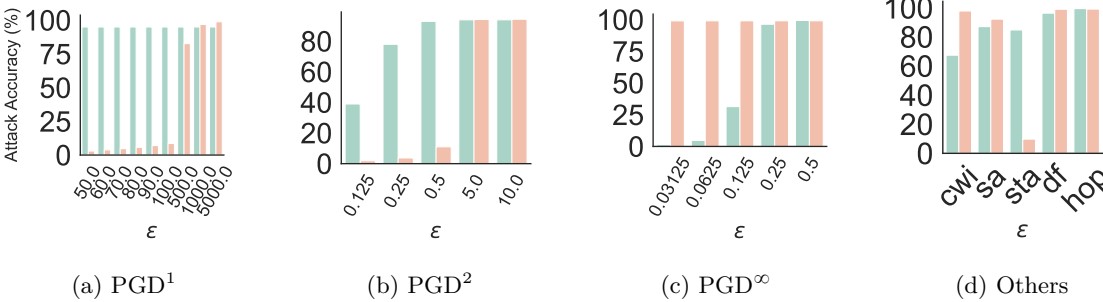

| (a) $PGD^1$ | (b) $PGD^2$ | (c) $PGD^\infty$ | (d) Others |

Figure 1: Percentage of successful attacks depending on the $L_p$-norm constraint, the maximal perturbation $\varepsilon$ and the attack algorithm on ResNet18 (green) and ViT (orange).

Given that the variety of attacks comes from choosing the $L_p$-norm constraint and the maximal allowed $L_p$-norm perturbation $\varepsilon$, it is crucial to select them carefully. Adversarial attacks and defense mechanisms have been widely studied for classical convolutional networks, particularly for ResNets, with an input size of $32 \times 32 \times 3$ (Goodfellow et al., 2014; Moosavi-Dezfooli et al., 2016; Zhang et al., 2019; Madry et al., 2018; Xu et al., 2018; Meng & Chen, 2017). Hence, when we change the input size, we need to choose the maximally allowed perturbation $\varepsilon$. It comes naturally to perform a comparison of the success rates of attacks between attacks on ResNets (with input size $32 \times 32 \times 3$) and ViTs (with input size $(224 \times 224 \times 3)$.

In Fig. 1, we present the success rates for attacks on Resnet18 (resp. on ViT-B) in blue (resp. in orange), for different attack mechanisms, different $L_p$-norms and different maximal perturbation $\varepsilon$ (the results for FGSM and BIM are relegated to Appendix E). Attacks behave differently depending on the input size: on $L_\infty$-norm constraints, at equal $\varepsilon$, the attacks are more potent on the ViT than on ResNet18. Indeed, the input of a ViT has more pixels than the input of a ResNet ($32 \times 32 \times 3$ for ResNet and $224 \times 224 \times 3$ for ViT). Limiting the perturbation by an $L_\infty$-norm constraint, i.e., controlling the maximal perturbation pixel-wise without controlling the number of modified pixels, will create samples further away from the original sample if it has more pixels. On the contrary, under $L_1$ and $L_2$-norm constraints, the opposite behavior is observable: at fixed $\varepsilon$, the attacks are more potent on ResNets with smaller inputs than on ViTs. This can be explained by the fact that limiting $L_1$ or $L_2$-norm perturbations controls the average perturbations on the whole input sample. The modifications are, therefore, smaller pixel-wise if the image is bigger. While on ResNets, the classical values of $\varepsilon$ are lower than 40 on $L_1$-norm constraints and 2 on $L_2$-norm constraints, we had to increase the maximum $\varepsilon$ studied for those $L_p$-norm constraint to have successful enough attacks. Finally, Spatial Transformation Attacks (STA) disturb ResNets' inputs functioning more easily than ViTs'.

**Summary.** To sum up, in the remaining of the paper, under $L_1$-norm constraint, we craft $PGD^1$ attacks with maximum norm constraint $\varepsilon \in \{50, 60, 70, 80, 90, 100, 500, 1000, 5000\}$. For the $L_2$-norm, we consider $PGD^2$ with $\varepsilon \in \{0.125, 0.25, 0.5, 5, 10\}$, DF with no $\varepsilon$, and HOP attacks with 3 restarts and $\varepsilon = 0.1$. Under

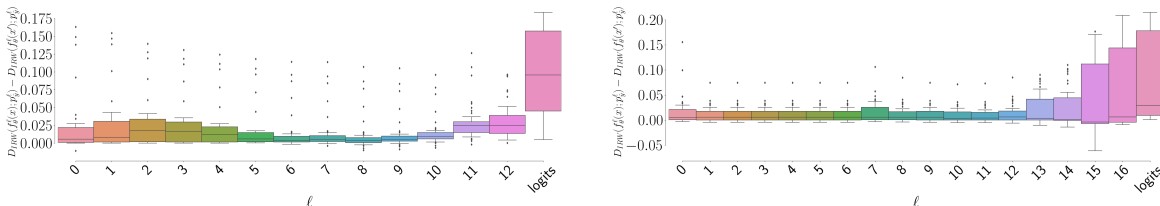

Figure 2: Difference between natural and adversarial IRW depth values as a function of the layer on ViT-B (left) and on ResNet50 (right), averaged over the attacks.

Table 3: Results on ViT-B averaged over the different attacks for each considered $L_p$-Norm constraints for `APPROVED`, FS and JTLA, along with the Averaged results over the norms tested. The results are presented as mean $\pm standard\_deviation$. The best results are presented in **bold**.

| | APPROVED | | | | | | FS | | | | | | JTLA | | | | | |
|---|---|---|---|---|---|---|---|---|---|---|---|---|---|---|---|---|---|---|
| | CIFAR10 | | CIFAR100 | | Tiny | | CIFAR10 | | CIFAR100 | | Tiny | | CIFAR10 | | CIFAR100 | | Tiny | |
| | AUROC↑ | FPR↓90% | AUROC↑ | FPR↓90% | AUROC↑ | FPR↓90% | AUROC↑ | FPR↓90% | AUROC↑ | FPR↓90% | AUROC↑ | FPR↓90% | AUROC↑ | FPR↓90% | AUROC↑ | FPR↓90% | AUROC↑ | FPR↓90% |
| $L_1$ | **94.0** ±5.2 | **13.2** ±13.5 | **78.3** ±7.6 | **46.4** ±13.1 | **75.2** ±1.3 | **59.2** ±2.7 | 79.5 ±3.3 | 34.9 ±8.3 | 71.1 ±5.1 | 55.5 ±8.0 | 54.2 ±14.0 | 75.1 ±11.6 | 78.9 ±12.5 | 61.9 ±20.4 | 68.9 ±12.0 | 67.0 ±14.6 | 68.8 ±1.5 | 67.4 ±3.1 |
| $L_2$ | **94.1** ±3.7 | **14.6** ±13.5 | **80.5** ±4.9 | **44.0** ±11.3 | **76.8** ±4.6 | **53.8** ±10.1 | 77.3 ±1.8 | 37.2 ±8.6 | 68.2 ±5.1 | 58.9 ±10.5 | 61.8 ±12.0 | 72.4 ±10.6 | 79.4 ±14.2 | 51.1 ±26.7 | 70.2 ±12.8 | 65.1 ±17.8 | 69.0 ±2.6 | 67.6 ±4.6 |
| $L_\infty$ | **95.3** ±6.5 | **13.4** ±20.9 | **86.7** ±0.4 | **29.9** ±19.3 | **91.8** ±8.8 | **19.1** ±19.6 | 73.0 ±3.5 | 53.4 ±18.3 | 62.6 ±6.8 | 67.3 ±11.6 | 74.6 ±17.8 | 61.2 ±23.9 | 70.1 ±14.6 | 66.7 ±27.9 | 62.7 ±13.7 | 74.9 ±16.2 | 65.2 ±6.1 | 72.5 ±9.2 |
| no Norm | **94.9** ±0.0 | **10.5** ±0.0 | **87.4** ±0.0 | **32.1** ±0.0 | **80.2** ±0.0 | **42.5** ±0.0 | 78.8 ±0.0 | 37.5 ±0.0 | 65.4 ±0.0 | 50.0 ±0.0 | 53.0 ±0.0 | 77.5 ±0.0 | 78.6 ±0.0 | 80.9 ±0.0 | 80.4 ±0.0 | 64.8 ±0.0 | 68.2 ±0.0 | 68.3 ±0.0 |
| Average | **94.7** ±5.6 | **13.5** ±17.5 | **83.2** ±8.9 | **37.2** ±17.8 | **83.9** ±10.3 | **37.7** ±23.8 | 75.8 ±4.2 | 44.2 ±16.4 | 66.1 ±7.0 | 62.0 ±11.6 | 65.8 ±18.0 | 67.7 ±19.6 | 74.6 ±14.4 | 62.6 ±26.3 | 66.4 ±13.6 | 70.5 ±16.5 | 67.0 ±4.9 | 70.0 ±7.4 |

$L_\infty$-norm constraint, we consider $PGD^\infty$, BIM and FGSM attacks with $\varepsilon \in \{0.03125, 0.0625, 0.125, 0.25, 0.5\}$, $CW^\infty$ with $\varepsilon = 0.3125$ and SA with $\varepsilon = 0.125$. Finally, STA attacks, which are not subject to a norm constraint, can rotate the input up to $60°$ and translate it up to 10 pixels.

### 4.3 Locating the relevant information

In the previous section, we saw that the attacks behave differently w.r.t. the classifier on which they are perpetrated. We continue this investigation by examining the differences between the two models from the depth scores' perspective. In this framework, we define the layer to have relevant information when the difference between the depth score on the natural and the depth score on the adversarial is significant. Indeed, the higher the difference, the more evident the shift between the distributions of the natural and the adversarial induced by the depth score will be, and hence the easier it will be to find a threshold that distinguishes natural from adversarial samples. We start by computing layer per layer the differences between the IRW depth on the natural samples $(D_{\text{IRW}}(f_\theta^\ell(x); p_{\hat{y}}^\ell))$ and on the adversarial samples $(D_{\text{IRW}}(f_\theta^\ell(x'); p_{\hat{y}}^\ell))$ both for ViT and for ResNet50. Both models have an input size of $224 \times 224 \times 3$. In Fig. 2, we plot the mean and standard deviation for each layer and each network. The diamond points represent the outliers. Fig. 2 shows that the information about whether a sample is natural or adversarial, based on the study of the IRW depth is significantly spread across the ResNet18 model: in each layer, the values range between 0 and 0.06. On the contrary, on ViT, this information is concentrated in the logit layer, where the values range between 0.05 and 0.2, while the values range from 0 to 0.05 for the other layers. To summarize, while relevant information to distinguish between natural and adversary samples is diffused in the ResNet18 model, which has small and similar values for all the layers, the most valuable information is instead concentrated at the logit layer for the ViT network, which experiences larger values only for that particular layer. It seems, therefore, relevant to build a detector *specific* for vision transformers based *only* on the output of the logit layer.

## 5 Experiments

### 5.1 Results

**Performances of `APPROVED` compared to other unsupervised detection methods.** In Tab. 10, Tab. 11, Tab. 12, Tab. 13, Tab. 14, and Tab. 15 relegated to Appendix H, we report the detailed results for each considered detection method under the threat of each attack mechanism, $L_p$-norm constraint and

Table 4: Results on ViT-L averaged over the different attacks for each considered $L_p$-Norm constraints for `APPROVED`, FS and JTLA, along with the Averaged results over the norms tested. The results are presented as mean $\pm standard\_deviation$. The best results are presented in **bold**.

| | APPROVED | | | | | | FS | | | | | | JTLA | | | | | |
|---|---|---|---|---|---|---|---|---|---|---|---|---|---|---|---|---|---|---|
| | CIFAR10 | | CIFAR100 | | Tiny | | CIFAR10 | | CIFAR100 | | Tiny | | CIFAR10 | | CIFAR100 | | Tiny | |
| | AUROC↑ | FPR↓90% | AUROC↑ | FPR↓90% | AUROC↑ | FPR↓90% | AUROC↑ | FPR↓90% | AUROC↑ | FPR↓90% | AUROC↑ | FPR↓90% | AUROC↑ | FPR↓90% | AUROC↑ | FPR↓90% | AUROC↑ | FPR↓90% |
| $L_1$ | **93.2** ±6.8 | **15.1** ±19.6 | 80.8 ±11.1 | 38.6 ±16.2 | 76.4 ±6.4 | 54.0 ±7.5 | 73.8 ±7.1 | 49.9 ±8.7 | 64.1 ±3.4 | 71.7 ±5.5 | 60.7 ±4.8 | 75.4 ±6.2 | 83.0 ±14.5 | 43.7 ±28.6 | 74.5 ±13.9 | 57.3 ±18.8 | 68.9 ±7.8 | 64.3 ±11.2 |
| $L_2$ | **93.2** ±4.3 | 18.3 ±15.5 | 81.5 ±9.6 | 37.9 ±15.7 | 76.5 ±8.3 | 52.9 ±14.0 | 71.8 ±6.6 | 50.8 ±6.2 | 62.7 ±4.7 | 71.9 ±4.6 | 59.3 ±4.8 | 77.0 ±4.9 | 82.6 ±14.7 | 47.7 ±26.1 | 77.0 ±4.9 | 59.8 ±19.5 | 68.7 ±7.4 | 66.9 ±11.1 |
| $L_\infty$ | **94.3** ±7.3 | **16.0** ±22.4 | 85.2 ±10.3 | 31.2 ±20.4 | 76.8 ±8.3 | 47.7 ±15.3 | 72.8 ±11.8 | 52.2 ±9.5 | 60.7 ±13.3 | 68.6 ±7.2 | 69.0 ±13.8 | 61.1 ±18.2 | 74.6 ±13.6 | 62.0 ±28.8 | 66.8 ±13.3 | 67.2 ±17.3 | 64.2 ±7.2 | 71.7 ±11.6 |
| no Norm | **94.3** ±0.0 | **12.6** ±0.0 | 89.4 ±0.0 | 26.5 ±0.0 | 85.7 ±0.0 | 30.7 ±0.0 | 69.5 ±0.0 | 52.9 ±0.0 | 59.2 ±0.0 | 71.1 ±0.0 | 63.4 ±0.0 | 69.3 ±0.0 | 90.3 ±0.0 | 34.8 ±0.0 | 83.3 ±0.0 | 45.5 ±0.0 | 74.1 ±0.0 | 52.4 ±0.0 |
| Average | **93.8** ±6.3 | **16.1** ±19.5 | 83.5 ±10.2 | 34.3 ±18.1 | 76.9 ±7.6 | 50.0 ±13.3 | 72.8 ±9.4 | 51.3 ±8.4 | 61.6 ±8.0 | 70.3 ±6.2 | 64.6 ±11.0 | 68.4 ±15.3 | 78.9 ±14.2 | 53.4 ±28.5 | 70.6 ±13.3 | 62.5 ±18.1 | 66.7 ±7.5 | 68.2 ±11.7 |

Table 5: Results on ViT-B averaged over the different types of attack mechanism for `APPROVED`, FS, and JTLA, along with the averaged results over the norms. The results are presented as mean $\pm standard\_deviation$. The best results are presented in **bold**. Dashed values (–) corresponds to attacks that take more than 48 hours to run on V100 GPUs.

| | APPROVED | | | | | | FS | | | | | | JTLA | | | | | |
|---|---|---|---|---|---|---|---|---|---|---|---|---|---|---|---|---|---|---|
| | CIFAR10 | | CIFAR100 | | Tiny | | CIFAR10 | | CIFAR100 | | Tiny | | CIFAR10 | | CIFAR100 | | Tiny | |
| | AUROC↑ | FPR↓90% | AUROC↑ | FPR↓90% | AUROC↑ | FPR↓90% | AUROC↑ | FPR↓90% | AUROC↑ | FPR↓90% | AUROC↑ | FPR↓90% | AUROC↑ | FPR↓90% | AUROC↑ | FPR↓90% | AUROC↑ | FPR↓90% |
| PGD | 95.5 ±4.3 | 9.6 ±10.8 | 81.3 ±7.8 | 41.2 ±14.6 | 81.0 ±10.2 | 45.0 ±24.1 | 77.2 ±3.8 | 44.4 ±14.2 | 70.1 ±4.9 | 62.5 ±11.1 | 65.6 ±19.0 | 66.2 ±22.5 | 72.9 ±14.3 | 67.0 ±22.0 | 64.1 ±13.3 | 71.6 ±16.7 | 66.4 ±3.9 | 70.9 ±5.8 |
| BIM | 96.8 ±4.4 | 7.1 ±10.1 | 82.1 ±13.1 | 37.9 ±26.2 | 95.0 ±7.4 | 11.8 ±16.6 | 71.2 ±1.5 | 69.6 ±2.9 | 64.3 ±1.9 | 77.8 ±3.4 | 86.5 ±2.2 | 60.4 ±19.2 | 58.6 ±2.9 | 84.2 ±3.1 | 51.6 ±0.6 | 86.5 ±0.7 | 60.9 ±2.5 | 79.3 ±3.6 |
| FGSM | 90.5 ±8.8 | 29.7 ±29.4 | 90.4 ±6.4 | 23.9 ±15.8 | 85.6 ±7.2 | 33.5 ±14.6 | 73.7 ±4.3 | 32.7 ±5.0 | 54.8 ±6.2 | 56.0 ±5.4 | 52.8 ±3.3 | 75.1 ±3.3 | 85.3 ±8.7 | 43.5 ±38.5 | 76.5 ±4.4 | 63.9 ±9.0 | 73.1 ±1.9 | 61.0 ±3.4 |
| HOP | 98.3 ±0.0 | 3.3 ±0.0 | 89.1 ±0.0 | 24.8 ±0.0 | 87.1 ±0.0 | 31.8 ±0.0 | 74.5 ±0.0 | 25.0 ±0.0 | 62.7 ±0.0 | 50.0 ±0.0 | 59.1 ±0.0 | 76.3 ±0.0 | 93.9 ±0.0 | 8.6 ±0.0 | 81.7 ±0.0 | 52.1 ±0.0 | 73.4 ±0.0 | 60.6 ±0.0 |
| DeepFool | 86.5 ±0.0 | 45.4 ±0.0 | 75.5 ±0.0 | 59.9 ±0.0 | – | – | 79.7 ±0.0 | 31.2 ±0.0 | 62.2 ±0.0 | 50.0 ±0.0 | – | – | 80.7 ±0.0 | 60.5 ±0.0 | 70.5 ±0.0 | 75.1 ±0.0 | – | – |
| SA | 98.2 ±0.0 | 3.3 ±0.0 | 89.6 ±0.0 | 26.0 ±0.0 | 77.0 ±0.0 | 49.1 ±0.0 | 72.0 ±0.0 | 25.0 ±0.0 | 63.3 ±0.0 | 50.0 ±0.0 | 48.7 ±0.0 | 78.5 ±0.0 | 93.0 ±0.0 | 13.6 ±0.0 | 87.7 ±0.0 | 30.8 ±0.0 | 70.6 ±0.0 | 63.0 ±0.0 |
| CW | 90.4 ±0.0 | 30.6 ±0.0 | 81.7 ±0.0 | 42.2 ±0.0 | – | – | 78.8 ±0.0 | 37.5 ±0.0 | 67.0 ±0.0 | 50.0 ±0.0 | – | – | 84.2 ±0.0 | 53.4 ±0.0 | 79.1 ±0.0 | 60.5 ±0.0 | – | – |
| STA | 94.9 ±0.0 | 10.5 ±0.0 | 87.4 ±0.0 | 32.1 ±0.0 | 80.2 ±0.0 | 42.5 ±0.0 | 78.8 ±0.0 | 37.5 ±0.0 | 65.4 ±0.0 | 50.0 ±0.0 | 53.0 ±0.0 | 77.5 ±0.0 | 78.6 ±0.0 | 80.9 ±0.0 | 80.4 ±0.0 | 64.8 ±0.0 | 68.2 ±0.0 | 68.3 ±0.0 |

Table 6: Results on ViT-L averaged over the different types of attack mechanism for `APPROVED`, FS, and JTLA, along with the averaged results over the norms. The results are presented as mean $\pm standard\_deviation$. The best results are presented in **bold**.

| | APPROVED | | | | | | FS | | | | | | JTLA | | | | | |
|---|---|---|---|---|---|---|---|---|---|---|---|---|---|---|---|---|---|---|
| | CIFAR10 | | CIFAR100 | | Tiny | | CIFAR10 | | CIFAR100 | | Tiny | | CIFAR10 | | CIFAR100 | | Tiny | |
| | AUROC↑ | FPR↓90% | AUROC↑ | FPR↓90% | AUROC↑ | FPR↓90% | AUROC↑ | FPR↓90% | AUROC↑ | FPR↓90% | AUROC↑ | FPR↓90% | AUROC↑ | FPR↓90% | AUROC↑ | FPR↓90% | AUROC↑ | FPR↓90% |
| PGD | 94.7 ±5.4 | 12.2 ±15.6 | 82.3 ±10.3 | 35.7 ±16.8 | 75.6 ±7.8 | 53.2 ±12.7 | 76.3 ±7.0 | 48.2 ±7.4 | 66.0 ±4.2 | 69.0 ±0.3 | 66.0 ±10.1 | 67.7 ±14.7 | 77.3 ±14.8 | 55.9 ±28.5 | 68.4 ±13.7 | 65.5 ±18.0 | 65.8 ±8.1 | 70.2 ±11.5 |
| BIM | 95.8 ±4.6 | 9.8 ±15.8 | 82.5 ±15.3 | 33.4 ±27.0 | 74.7 ±11.3 | 50.3 ±7 | 79.9 ±2.2 | 48.0 ±3.7 | 68.3 ±3.9 | 65.5 ±5.2 | 78.1 ±7.4 | 49.4 ±12.0 | 63.7 ±2.4 | 81.8 ±3.8 | 57.0 ±3.5 | 79.4 ±5.1 | 59.8 ±5.6 | 78.4 ±6.1 |
| FGSM | 88.8 ±9.8 | 33.6 ±30.5 | 85.9 ±7.3 | 33.2 ±20.1 | 79.4 ±1.9 | 45.6 ±4.7 | 56.9 ±5.8 | 65.3 ±4.6 | 44.2 ±11.1 | 76.3 ±2.8 | 53.9 ±2.1 | 79.2 ±2.2 | 88.3 ±8.2 | 38.0 ±28.5 | 80.1 ±6.2 | 49.2 ±15.7 | 71.1 ±2.6 | 60.7 ±8.7 |
| HOP | 98.2 ±0.0 | 3.6 ±0.0 | 91.2 ±0.0 | 19.7 ±0.0 | 85.3 ±0.0 | 32.3 ±0.0 | 68.2 ±0.0 | 48.4 ±0.0 | 55.0 ±0.0 | 74.5 ±0.0 | 52.6 ±0.0 | 81.7 ±0.0 | 93.4 ±0.0 | 25.7 ±0.0 | 86.4 ±0.0 | 44.7 ±0.0 | 73.0 ±0.0 | 54.4 ±0.0 |
| DeepFool | 88.0 ±0.0 | 42.2 ±0.0 | ± | ± | 80.4 ±0.0 | 47.4 ±0.0 | 64.0 ±0.0 | 58.2 ±0.0 | ± | ± | 57.6 ±0.0 | 80.0 ±0.0 | 87.6 ±0.0 | 44.9 ±0.0 | ± | ± | 72.5 ±0.0 | 61.1 ±0.0 |
| SA | 97.6 ±0.0 | 4.0 ±0.0 | 91.2 ±0.0 | 21.0 ±0.0 | 82.3 ±0.0 | 40.2 ±0.0 | 70.3 ±0.0 | 50.3 ±0.0 | 56.3 ±0.0 | 73.7 ±0.0 | 56.3 ±0.0 | 82.1 ±0.0 | 96.0 ±0.0 | 6.7 ±0.0 | 86.0 ±0.0 | 42.6 ±0.0 | 75.0 ±0.0 | 50.6 ±0.0 |
| CW | 88.4 ±0.0 | 38.1 ±0.0 | 77.0 ±0.0 | 53.9 ±0.0 | 73.7 ±0.0 | 57.6 ±0.0 | 67.8 ±0.0 | 53.1 ±0.0 | 55.8 ±0.0 | 76.4 ±0.0 | 51.9 ±0.0 | 83.9 ±0.0 | 87.3 ±0.0 | 43.6 ±0.0 | 74.5 ±0.0 | 64.2 ±0.0 | 66.7 ±0.0 | 70.4 ±0.0 |
| STA | 94.3 ±0.0 | 12.6 ±0.0 | 89.4 ±0.0 | 26.5 ±0.0 | 85.7 ±0.0 | 30.7 ±0.0 | 69.5 ±0.0 | 52.9 ±0.0 | 59.2 ±0.0 | 71.1 ±0.0 | 63.4 ±0.0 | 69.3 ±0.0 | 90.3 ±0.0 | 34.8 ±0.0 | 83.3 ±0.0 | 45.5 ±0.0 | 85.7 ±0.0 | 30.7 ±0.0 |

maximum perturbation $\varepsilon$. In Tab. 3 and Tab. 4, we report the averaged AUROC↑ and FPR↓90% on each of the considered $L_p$-norm, along with the global average for each detector on CIFAR10, CIFAR100, and Tiny ImageNet on the ViT-B and ViT-L, respectively. Overall, `APPROVED` shows better results than the SOTA detection methods. On CIFAR10, `APPROVED` creates an increase of AUROC↑ of 18.9% (resp. 14.9%) and a decrease of FPR↓90% of 30.7% (resp. 37.3%) compared to the best performing state-of-the-art detector, i.e., FS (resp. JTLA) on ViT-B (resp. ViT-L). On CIFAR100, the improvements are 16.9% (resp. 12.9%) and 24.8% (resp. 28.2), respectively, while they are 16.9% (resp. 10.2%) and 32.3% (resp. 18.2%) on Tiny ImageNet. In addition, all methods have similar dispersions. Moreover, under specific $L_p$-norm constraints, our method consistently outperforms SOTA methods, especially under the $L_\infty$-norm constraint where `APPROVED` outperforms FS (resp. JTLA) by 22.3% (resp. 15.1%) in terms of AUROC↑ and 40.0% (resp. 48.7%) in terms of FPR↓90% on CIFAR10 when applied to ViT-B models. On ViT-L, `APPROVED` increases AUROC↑ values by 19.4% (resp. 10.2%) and decreases FPR↓90% values by 34.8 % (resp. 28.6%) compared to FS (resp. JTLA) on CIFAR10. Finally, by looking at the detailed results presented in Appendix H, we can deduce that FS and `APPROVED` have opposite behaviors: when the performances of FS decrease, `APPROVED`'s performances tend to improve. For example, under the $L_\infty$-norm constraint, `APPROVED` has more trouble detecting attacks with small perturbations, while FS has more difficulty detecting attacks with large perturbations. Indeed, since `APPROVED` measures the depth of a sample within a distribution, it will be able to recognize the strongest attacks well.

**Performances per attack.** In Tab. 5 and Tab. 6, we give the overall idea of the results on all three datasets per attack mechanism by showing them in terms of *mean* and *standard deviation* (std) on the AUROC↑ and the FPR↓90%, on each considered model. `APPROVED` turns out to consistently outperform the state-of-the-art

detectors for all datasets. In particular, we notice that the FGSM attacks that are the easiest to generate are the ones that present the highest diversity among the results in the methods examined. Indeed, by looking at Tab. 5, we can find larger values of the standard deviation in correspondence to that attack. Moreover, APPROVED can recognize attacks that are more difficult for the competitors (e.g., BIM for JTLA or FGSM for FS). We also observe that the most challenging task for APPROVED is to distinguish natural and adversarial samples when crafted with DeepFool and the Carlini&Wagner attack. However, it is the best choice even in this case as it reaches better performances than the other detectors.

## 5.2 Adaptive Attacks

In this experiment, we evaluate APPROVED against adaptive attacks, which has knowledge about the defense (Athalye et al., 2018; Tramer et al., 2020; Carlini & Wagner, 2017b). Two scenarios can be considered with adaptive attacks: whitebox and blackbox. Whitebox attacks (*e.g.,* BPDA, Athalye et al., 2018) are not straightforward to adapt in our case since finding a differentiable surrogate of IRW remains a very challenging open research question in the statistical community, which has never been tackled. As a matter of fact, the only attempt to approximate a non-differentiable depth was performed not on the IRW depth but on the Tukey depth in Dyckerhoff et al. (2021), with very poor results as pointed out in She et al. (2021). Thus, in this experiment, we rely on blackbox attacks and present the results in Fig. 3. We attacked both APPROVED and FS using a modified version of SA (Andriushchenko et al., 2020), for which the attack objective has been modified to allow the attacker to fool both the detection method as well as the classifier. We rely on a hyperparameter $\alpha$ that weighs the relative importance of the two parts of the objective.
*Remark.* Both methods have their advantages and drawbacks. Due to speed, memory requirements, and results on CIFAR10 when attacking a ViT-B, we decided to compare APPROVED to FS under adaptive attacks and discard JTLA. (see Appendix D.2). It is straightforward (cf. Fig. 3) that APPROVED is less sensitive to adaptive attacks than FS. These results further validate the use of the IRW depth to craft a detection method and further assesses the superiority of APPROVED. In Appendix F, we performed another adaptive attacks, based on a surrogate to the depth score.

## 5.3 Finer Analysis

**Per class analysis.** As explained in Sec. 3.2, APPROVED is based on the IRW depth, which computes the depth score of the sample w.r.t. the original distribution by class. Fig. 4 shows the per-class performances averaged over the different attacks on CIFAR10, while Fig. 8, relegated to Appendix G due to space constraints, shows the performances on CIFAR100. It is clear from Fig. 4 that APPROVED does not have equal performances in every class. In particular, some classes present extremely high mean average AUROC↑ (i.e., class 7), and others exhibit very low FPR↓$_{90\%}$ (i.e., class 6). In contrast, some others have their adversarial and clean samples tough to distinguish (i.e., classes 3 and 5). The same behavior is observable in CIFAR100 (see Fig. 8).

**AUROC↑ vs FPR↓$_{90\%}$.** We conclude our analysis by examining the trade-off between AUROC↑ and FPR↓$_{90\%}$ (see Fig. 5). The ideal method would concentrate the results on the upper left of the figure, which corresponds to high AUROC↑ and low FPR↓$_{90\%}$, while a poor detector would concentrate them in the bottom right corner of the figure, which corresponds to low AUROC↑ and high FPR↓$_{90\%}$. We observe that on CIFAR10, the APPROVED points are more concentrated in the upper left corner, while the FS points are concentrated in the center of the figure, and JTLA's are spread across the entire figure. On CIFAR100 and

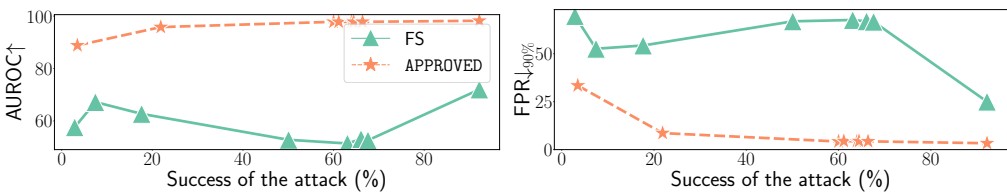

Figure 3: Detector Performances under blackbox Adaptive Attack.

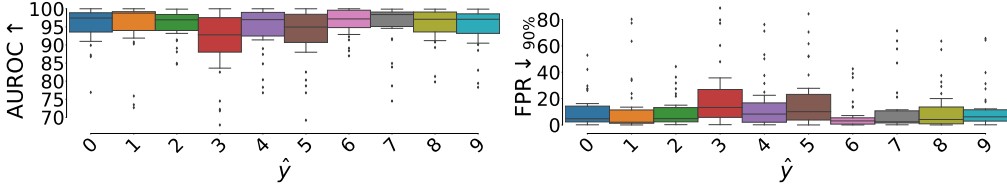

Figure 4: `APPROVED`'s AUROC↑ and FPR↓$_{90\%}$ per class, averaged over CIFAR10.

Tiny ImageNet, the results for our method are slightly more spread out in the top left and center of the figure, while for FS, they are still in the center and spread out in the center and lowest right corner for JTLA. Note that FS has a different behavior than expected, i.e., the line connecting the top left corner with the bottom right corner. This behavior change can be observed for FPR↓$_{90\%}$ between 25%-35% on CIFAR10 and between 50%-75% on CIFAR100 and Tiny ImageNet. On CIFAR10, FS presents a lower AUROC↑ for a fixed FPR↓$_{90\%}$ than expected, whereas, on CIFAR100, it presents a lower AUROC↑ (for FPR↓$_{90\%}$ values between 50%-60%) or higher (for FPR↓$_{90\%}$ values around 75%) than expected.

## 6 Conclusions and Limitations

We introduced `APPROVED`, an efficient unsupervised detection method designed to defend against adversarial attacks. In contrast with previous detection methods built for ResNet architectures, `APPROVED` is well suited for vision transformers which now represent the state-of-the-art. While the relevant information about the discrepancy between clean and adversarial samples is distributed across all layers of ResNets, for the transformers, it was empirically shown to be concentrated in the logit layer. This motivated us to build `APPROVED` on top of this logit layer by computing a similarity score between an input sample and the training distribution based on the statistical notion of *data depth*. We chose to use the Integrated Rank-Weighted depth, which lends itself to fast inference computations and is non-differentiable, making it harder for gradient-based adversarial methods to craft malicious samples.We conduct extensive numerical experiments and prove that `APPROVED` significantly outperforms the other state-of-the-art methods.

**Future Research.** We think our method paves the way for future research efforts. Indeed, there is still room for improvement: even if the AUROC↑ performances are good, the FPR↓$_{90\%}$ are also fairly high. We believe the idea of leveraging information contained in layers of transformers through data depths can be fruitful in improving defense mechanisms against adversarial attacks. Furthermore, It would be interesting to investigate other architectures to find whether a specific layer contains the relevant information, as for ViTs, or if the information is spread across the network, as for ResNets. Our research is expected to have a positive societal impact by protecting the integrity of AI systems, especially necessary in critical systems such as autonomous cars (Morgulis et al., 2019) or stock predictions (Xie et al., 2022).

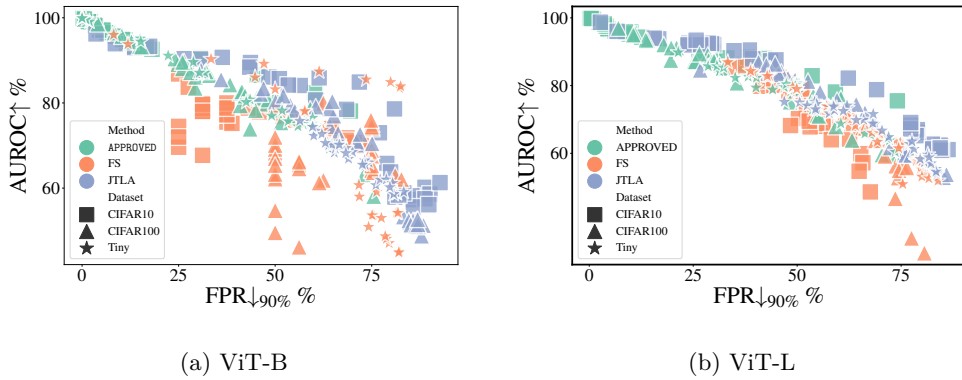

(a) ViT-B

(b) ViT-L

Figure 5: AUROC↑ as a function of FPR↓$_{90\%}$ for `APPROVED`, FS, and JTLA on all considered datasets.

**Broader Impact Statement**

We believe our work will have a positive impact on society. Indeed, in this work, we propose a method to improve Deep Learning systems to improve our reliability in Deep Neural Networks, as their potential failure has been raising many concerns, limiting their adoption in critical applications.

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

## A  Training and Evaluation Details

**Training.**  We compare the different detection methods on three vision datasets: CIFAR10, CI-FAR100 (Krizhevsky et al.) and Tiny ImageNet (Jiao et al., 2019) for which we use the ViT models presented in Sec. 4.1 to build a classifier.

We trained two different models: a ViT and a ResNet18. The ResNet18 has been trained on 100 epochs, with a Stochastic Gradient Descent (SGD) optimizer, with a learning rate of 0.1, a momentum of 0.9, and a weight decay of $10^{-5}$. We use the base model with 16 layers (85.8 million of parameters) from `https://github.com/jeonsworld/ViT-pytorch` trained on ImageNet (Deng et al., 2009) as our ViT classifier for CIFAR10 and CIFAR100. To train it, we set the batch size to 512. The learning rate of SGD (Ruder, 2016) is set to $3 \times 10^{-2}$ and we use 500 warming steps with no gradient accumulation (Vaswani et al., 2017). For Tiny ImageNet, we used a ViT with 16 layers as the underlying classifier, trained by Huynh (2022) and available at `https://github.com/ehuynh1106/TinyImageNet-Transformers`. Note that we only use the class token to output the layer-wise input's representations.

*Remark.* We compare our proposed `APPROVED` method with FS and JTLA, recalled in Sec. 2.2. We train JTLA according to its original training procedure, while FS and our `APPROVED`, presented in Sec. 3.2, do not require any training.

**Evaluation.** To compute the IRW depth, we set $n_{proj}$ to the standard value of 10000 ($\approx$ ten times the dimension), as advised in Staerman et al. (2021a). To generate the attacks, we used the Cross-Entropy loss, we set the number of steps to 100 and the step size to 0.01 for PGD$^{\infty}$ and BIM, 0.1 for PGD$^2$ and 5 for small $\varepsilon$ values and 50 for larger one for PGD$^1$. We used 5 restarts.

## B  Approximation Algorithm

This appendix displays the algorithm used to compute the IRW depth (see Algo. 1).

---
**Algorithm 1** Approximation of the IRW depth

---
*Initialization:* test sample $x$, $n_{\mathrm{proj}}$, $\mathbf{X} = [x_1, \ldots, x_n]^{\top}$.

    Construct $\mathbf{U} \in \mathbb{R}^{d \times n_{\mathrm{proj}}}$ by sampling uniformly $n_{\mathrm{proj}}$ vectors $U_1, \ldots, U_{n_{\mathrm{proj}}}$ in $\mathbb{S}^{d-1}$

    Compute $\mathbf{M} = \mathbf{X}\mathbf{U}$ and $x^{\top}\mathbf{U}$

    Compute the rank value $\sigma(j)$, the rank of $x^{\top}\mathbf{U}$ in $\mathbf{M}_{:,j}$ for every $j \leqslant n_{\mathrm{proj}}$

    Set $D = \frac{1}{n_{\mathrm{proj}}} \sum_{j=1}^{n_{\mathrm{proj}}} \sigma(j)$

    **Output**: $\widetilde{D}_{\mathrm{IRW}}^{\mathrm{MC}}(x, \mathbf{X}) = D$

---

**Complexity.** The complexity of the algorithm is detailed as follows. Line 1 requires sampling $n_{\mathrm{proj}}$ Gaussian samples and normalizing them in order to define unit sphere directions and can be computed in $\mathcal{O}(n_{\mathrm{proj}}d)$. Line 2 requires $\mathcal{O}(n_{\mathrm{proj}}dn)$ to project data on the $n_{\mathrm{proj}}$ unit sphere Monte-Carlo directions. Line 3 requires computing the sorting operation on $n_{\mathrm{proj}}$ columns of the matrix $M$ and then leads to a complexity of $\mathcal{O}(n_{\mathrm{proj}}n)$. Line 4 requires the computation of the mean and can be done in $n_{\mathrm{proj}}$ operations. Finally, the total complexity of the algorithm is then in $\mathcal{O}(n_{\mathrm{proj}}dn)$, which is linear in all of its parameters.

**Remarks.** Given that the algorithm is linear in all its parameters, computing the IRW depth can be scaled to any dataset. Note that the IRW data depth makes no assumption on the training distribution. In line 3 of Algo. 1, "rank values" consists in ranking the elements of the projection of each input on U. This is achieved by a sorting algorithm. This step allows us to define an ordering of the projected inputs, which is used to compute the final depth score.

## C   Intuition on Data Depth

To provide a better sense of why data-depths are useful tools to distinguish between natural and attacked samples, we decided to provide a graphical explanation.

On CIFAR-10, using a ViT-B_16 as the underlying classifier, we plotted the training, natural and adversarial logits for class 0 and class 1 output by the network, using a t-SNE to reduce the dimensions to 2. The visualisation is provided in Fig. 6.

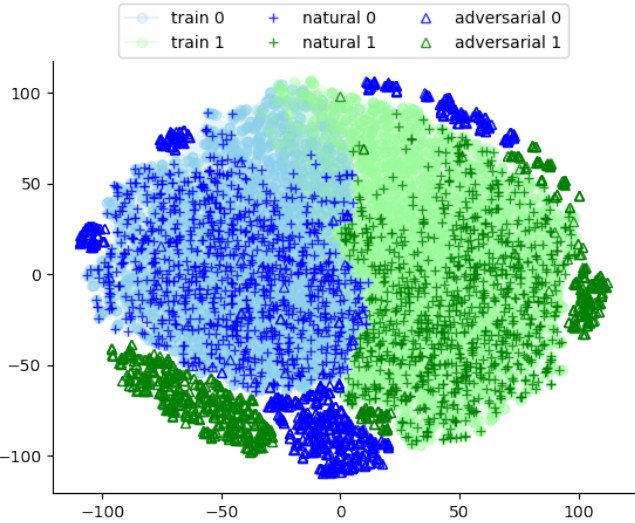

Figure 6: 2D representation of the logits distribution for class 0 (blue) and class 1 (green). The training samples are in light colors, using circles (○), the natural testing samples are represented by crosses (+) and the adversarial samples are triangles (△).

From Fig. 6, we observe that the natural testing samples are deep inside the training distribution while the adversarial samples are on the border. Data depth, which computes the distance between a reference distribution and new samples, seems to be a good fit to distinguish between clean and attacked samples.

# D   Time and Computational Requirements

## D.1   To generate attacks

We here present the computational requirements to generate the attacks on the transformer, along with the required time to generate them. We use the Adversarial-Robustness Toolbox (ART; Nicolae et al., 2018) to generate the attacks.

Table 7: Resources and time needed to generate different types of attack on CIFAR10.

| Attack | GPUs | CPUs | Time |
|--------|------|------|------|
| FGSM | V100-32G | 20G | 0h25 |
| BIM | V100-32G | 20G | 3h13 |
| PGD | V100-32G | 20G | 4h30 |
| DF | V100-32G | 20G | 1h54 |
| HOP | V100-32G | 20G | 47h39 |
| CW$^\infty$ | V100-32G | 30G | 2h48 |
| SA | V100-32G | 20G | 5h04 |
| STA | V100-32G | 20G | 1h25 |

## D.2   To deploy detectors

This section presents the computational requirements, along with the time needed to deploy each of the studied detection methods on a ViT-B on CIFAR10. We use the codes available at https://github.com/aldahdooh/detectors_review for FS. For JTLA, we used the code proposed by the authors, available at https://github.com/jayaram-r/adversarial-detection.

Table 8: Resources and time needed to train and test each detection method.

| Method | GPUs | CPUs | Training Time | Testing Time |
|--------|------|------|---------------|--------------|
| APPROVED | V100-32G | 40G | N/A | 0h11 |
| FS | V100-32G | 80G | N/A | 0h53 |
| JTLA | V100-32G | 180G | 0h42 | 1h26 |

# E   Success Rates of Attacks on CIFAR10

We here report the success rate per attack for all the different threat mechanisms (i.e., $PGD^1$, $PGD^2$, $PGD^\infty$, BIM, FGSM, $CW^\infty$, SA, STA, DF and HOP) on a ViT-B on CIFAR10. In orange are the attack performances on ViT, while the ones on ResNet are in green (see Sec. 4 for a detailed analysis).

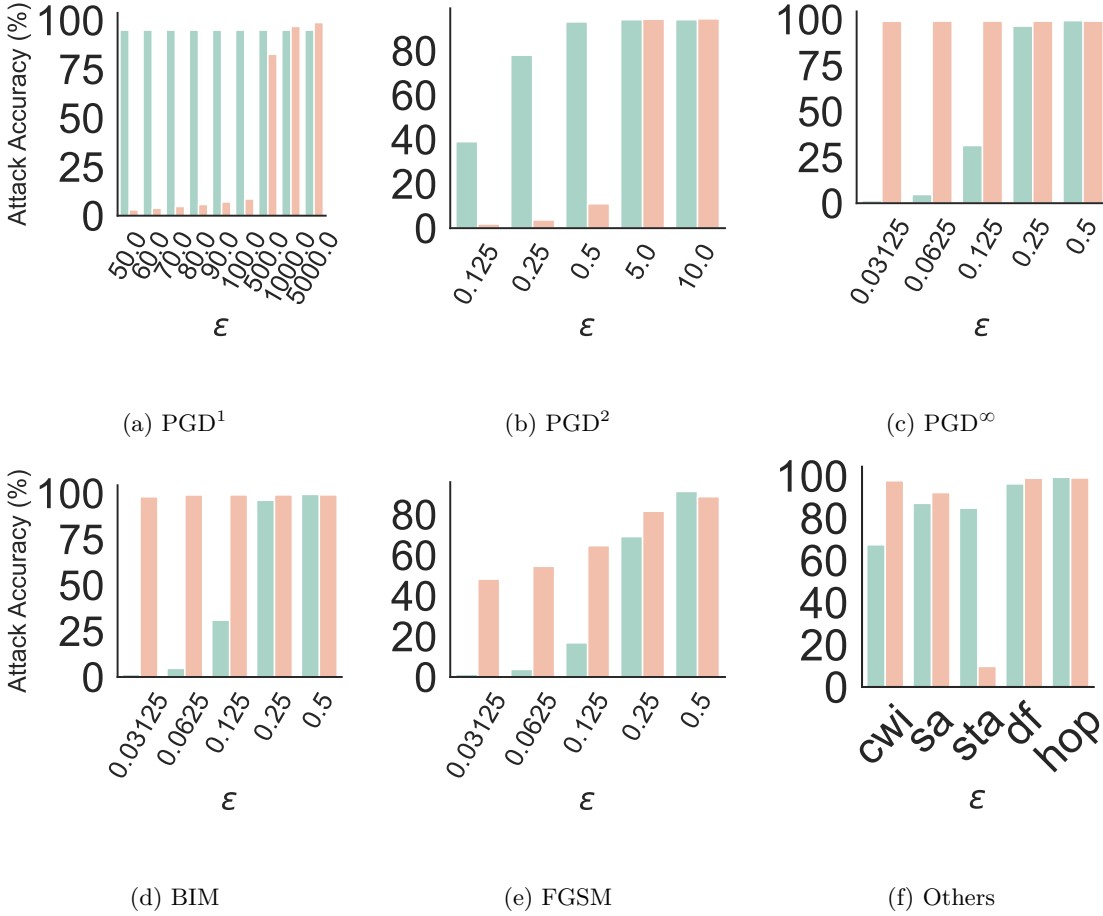

(a) $PGD^1$            (b) $PGD^2$            (c) $PGD^\infty$

(d) BIM            (e) FGSM            (f) Others

Figure 7: Percentage of successful attacks on a ViT-B depending on the $L_p$-norm constraint, the maximal perturbation $\varepsilon$ and the attack algorithm on ResNet18 (vert) and ViT-B (orange).

## F  Adaptive Attacks

To further analyse our proposed method, we investigate a surrogate of the depth score.

As the depth compute the similarity between a new point and a reference distribution, we could think of the following surrogate:

$$\underset{\|x-x'\|_p < \varepsilon}{\arg\min} \|f^{L-1}(x') - f^{L-1}(\widehat{x})\|_2^2, \tag{4}$$

where $\widehat{x}$ is a randomly selected training sample with $\widehat{y} \neq y$.

We used the PGD algorithm to test this new surrogate on CIFAR10 using the ViT-B, with $\varepsilon$ varying between 0.03125 and 0.5. The results are reported in Table 9.

| Adaptive Attacks - Surrogate | | |
|---|---|---|
| Norm $L_\infty$ | AUROC↑ | FPR↓$_{90\%}$ |
| PGD$^\infty$ | | |
| $\varepsilon = 0.03125$ | 60.0 | 86.9 |
| $\varepsilon = 0.0625$ | 57.6 | 88.1 |
| $\varepsilon = 0.125$ | 56.9 | 88.3 |
| $\varepsilon = 0.25$ | 56.8 | 88.2 |
| $\varepsilon = 0.5$ | 56.8 | 88.3 |

Table 9: AUROC↑ and FPR↓$_{90\%}$ of APPROVED for the adaptive attack using the surrogate in Equation 4 using the PGD algorithm, varying $\varepsilon$

Although its performance drops a little under this specific attack, our proposed method does not collapse. As a matter of fact, it achieves similar results as JTLA under non-adaptive attacks.

## G   Per Class Analysis

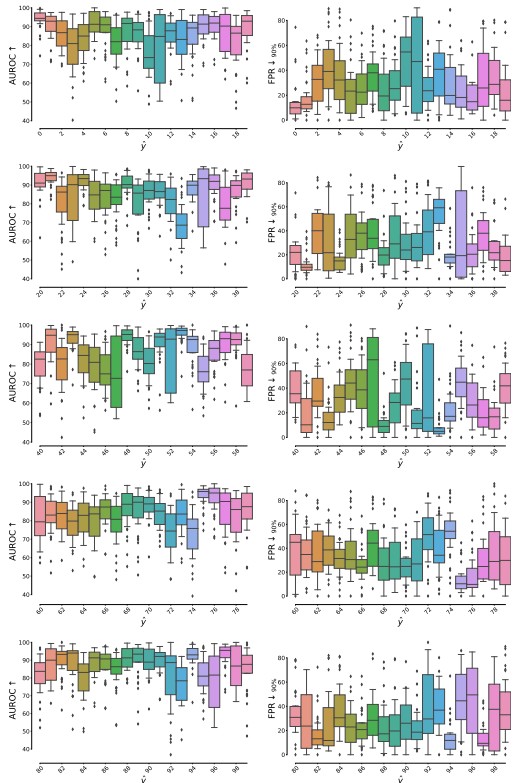

Figure 8: APPROVED's AUROC↑ and FPR↓$_{90\%}$ per class, averaged over the attacks on CIFAR100 on a ViT-B.

As for CIFAR10 (see Sec. 5), the detector performances depend on the predicted class. Some classes are easy to detect (i.e., classes 0, 21, 53, 75, and 94), while others are more difficult (i.e., classes 3, 10, 33, 47, 60, 74, and 93). Some have low variance (i.e., 0, 1, 24, 34, 75, 82, and 94), while others have a substantial dispersion (i.e., 11, 35,47, 52, 96, and 98).

# H    Detailed results for CIFAR10, CIFAR100, and Tiny ImageNet

Table 10: AUROC↑ and FPR↓$_{90\%}$ for each considered attack mechanisms, L$_p$-norm constraint and $\varepsilon$ on CIFAR10 for APPROVED, FS, and JTLA on a ViT-B. The best result for each attack is shown in **bold**.

| CIFAR10 - ViT-B | | | | | | |
|---|---|---|---|---|---|---|
| Norm L1 | APPROVED | | FS | | JTLA | |
| | AUROC↑ | FPR↓$_{90\%}$ | AUROC↑ | FPR↓$_{90\%}$ | AUROC↑ | FPR↓$_{90\%}$ |
| PGD[1] | | | | | | |
| $\varepsilon = 50$ | **97.2** | **5.0** | 77.6 | 37.5 | 89.5 | 43.6 |
| $\varepsilon = 60$ | **97.0** | **5.7** | 77.4 | 37.5 | 90.3 | 30.7 |
| $\varepsilon = 70$ | **96.4** | **6.8** | 78.0 | 31.2 | 88.4 | 43.4 |
| $\varepsilon = 80$ | **95.7** | **8.6** | 78.1 | 31.2 | 85.9 | 61.4 |
| $\varepsilon = 90$ | **94.8** | **11.1** | 78.7 | 31.2 | 84.9 | 71.8 |
| $\varepsilon = 100$ | **93.9** | **13.9** | 79.0 | 37.5 | 86.1 | 48.4 |
| $\varepsilon = 500$ | 80.1 | 50.1 | **86.8** | **25.0** | 65.8 | 78.9 |
| $\varepsilon = 1000$ | **93.0** | **14.2** | 83.7 | 37.5 | 60.2 | 88.8 |
| $\varepsilon = 5000$ | **98.0** | **3.6** | 76.0 | 55.2 | 58.6 | 90.1 |
| Norm L2 | APPROVED | | FS | | JTLA | |
| | AUROC↑ | FPR↓$_{90\%}$ | AUROC↑ | FPR↓$_{90\%}$ | AUROC↑ | FPR↓$_{90\%}$ |
| PGD[2] | | | | | | |
| $\varepsilon = 0.125$ | **97.1** | **4.5** | 75.5 | 37.5 | 90.7 | 36.3 |
| $\varepsilon = 0.25$ | **97.1** | **5.5** | 77.2 | 37.5 | 90.4 | 26.4 |
| $\varepsilon = 0.5$ | **92.6** | **18.1** | 79.8 | 31.2 | 84.1 | 56.5 |
| $\varepsilon = 5$ | **93.3** | **13.6** | 77.0 | 45.9 | 58.5 | 83.5 |
| $\varepsilon = 10$ | **94.1** | **11.5** | 76.8 | 52.1 | 57.3 | 85.7 |
| HOP | | | | | | |
| $\varepsilon = 0.1$ | **98.3** | **3.3** | 74.5 | 25.0 | 93.9 | 8.6 |
| DeepFool | | | | | | |
| No $\varepsilon$ | **86.5** | 45.4 | 79.7 | **31.2** | 80.7 | 60.5 |
| Norm L$_\infty$ | APPROVED | | FS | | JTLA | |
| | AUROC↑ | FPR↓$_{90\%}$ | AUROC↑ | FPR↓$_{90\%}$ | AUROC↑ | FPR↓$_{90\%}$ |
| PGD$^\infty$ | | | | | | |
| $\varepsilon = 0.03125$ | **96.5** | **6.4** | 78.7 | 42.9 | 58.2 | 85.4 |
| $\varepsilon = 0.0625$ | **99.1** | **2.1** | 73.4 | 64.7 | 57.5 | 88.8 |
| $\varepsilon = 0.125$ | **99.7** | **0.8** | 71.8 | 68.6 | 60.1 | 81.1 |
| $\varepsilon = 0.25$ | **99.8** | **0.5** | 70.9 | 70.0 | 57.9 | 81.0 |
| $\varepsilon = 0.5$ | **99.8** | **0.5** | 70.8 | 70.1 | 61.3 | 92.7 |
| BIM | | | | | | |
| $\varepsilon = 0.03125$ | **88.3** | **27.0** | 74.0 | 64.5 | 55.6 | 85.8 |
| $\varepsilon = 0.0625$ | **97.1** | **5.4** | 70.2 | 72.3 | 56.1 | 89.8 |
| $\varepsilon = 0.125$ | **99.0** | **2.2** | 70.0 | 72.2 | 58.9 | 83.3 |
| $\varepsilon = 0.25$ | **99.7** | **0.7** | 70.7 | 70.5 | 58.4 | 82.0 |
| $\varepsilon = 0.5$ | **99.9** | **0.2** | 71.2 | 68.4 | 63.8 | 79.9 |
| FGSM | | | | | | |
| $\varepsilon = 0.03125$ | 78.1 | 69.5 | 75.2 | **38.8** | 73.0 | 77.0 |
| $\varepsilon = 0.0625$ | 82.4 | 60.2 | 77.2 | **37.5** | 78.5 | 68.5 |
| $\varepsilon = 0.125$ | **93.1** | **16.6** | 78.9 | 31.2 | 85.7 | 51.1 |
| $\varepsilon = 0.25$ | **99.1** | **1.6** | 69.6 | 25.0 | 93.3 | 17.5 |
| $\varepsilon = 0.5$ | **99.7** | **0.6** | 67.7 | 31.2 | 96.2 | 3.6 |
| SA | | | | | | |
| $\varepsilon = 0.125$ | **98.2** | **3.3** | 72.0 | 25.0 | 93.0 | 13.6 |
| CW$^\infty$ | | | | | | |
| $\varepsilon = 0.3125$ | **90.4** | **30.6** | 78.8 | 37.5 | 84.2 | 53.4 |
| No Norm | APPROVED | | FS | | JTLA | |
| | AUROC↑ | FPR↓$_{90\%}$ | AUROC↑ | FPR↓$_{90\%}$ | AUROC↑ | FPR↓$_{90\%}$ |
| STA | | | | | | |
| No $\varepsilon$ | **94.9** | **10.5** | 78.8 | 37.5 | 78.6 | 80.9 |

Table 11: AUROC↑ and FPR↓$_{90\%}$ for each considered attack mechanisms, $L_p$-norm constraint and $\varepsilon$ on CIFAR10 for `APPROVED`, FS, and JTLA on a ViT-L. The best result for each attack is shown in **bold**.

| CIFAR10 - ViT-L | | | | | | |
|---|---|---|---|---|---|---|
| Norm L1 | APPROVED | | FS | | JTLA | |
| | AUROC↑ | FPR↓$_{90\%}$ | AUROC↑ | FPR↓$_{90\%}$ | AUROC↑ | FPR↓$_{90\%}$ |
| PGD[1] | | | | | | |
| $\varepsilon = 50$ | **96.2** | **5.6** | 67.9 | 57.1 | 92.1 | 29.9 |
| $\varepsilon = 60$ | **96.4** | **5.1** | 68.5 | 54.6 | 92.5 | 23.9 |
| $\varepsilon = 70$ | **96.6** | **4.7** | 68.8 | 56.3 | 94.0 | 15.6 |
| $\varepsilon = 80$ | **96.6** | **6.1** | 69.3 | 55.5 | 92.4 | 29.4 |
| $\varepsilon = 90$ | **96.5** | **5.6** | 70.1 | 55.4 | 92.3 | 26.8 |
| $\varepsilon = 100$ | **96.2** | **6.3** | 70.3 | 54.1 | 92.0 | 23.7 |
| $\varepsilon = 500$ | 78.1 | 59.2 | **79.8** | **41.1** | 69.1 | 77.3 |
| $\varepsilon = 1000$ | **86.4** | 37.4 | 85.4 | **33.6** | 61.3 | 82.6 |
| $\varepsilon = 5000$ | **96.2** | **6.1** | 83.5 | 41.4 | 61.4 | 84.4 |
| Norm L2 | APPROVED | | FS | | JTLA | |
| | AUROC↑ | FPR↓$_{90\%}$ | AUROC↑ | FPR↓$_{90\%}$ | AUROC↑ | FPR↓$_{90\%}$ |
| PGD[2] | | | | | | |
| $\varepsilon = 0.125$ | **95.2** | **9.6** | 68.0 | 56.2 | 92.5 | 28.6 |
| $\varepsilon = 0.25$ | **96.8** | **4.4** | 68.8 | 54.7 | 92.0 | 26.8 |
| $\varepsilon = 0.5$ | **95.8** | **8.7** | 72.2 | 52.2 | 90.5 | 38.5 |
| $\varepsilon = 5$ | **88.2** | **33.1** | 81.4 | 41.6 | 60.9 | 84.7 |
| $\varepsilon = 10$ | **90.2** | **26.4** | 80.1 | 44.4 | 61.6 | 84.4 |
| HOP | | | | | | |
| $\varepsilon = 0.1$ | **98.2** | **3.6** | 68.2 | 48.4 | 93.4 | 25.7 |
| DeepFool | | | | | | |
| No $\varepsilon$ | **88.0** | **42.2** | 64.0 | 58.2 | 87.6 | 44.9 |
| Norm L$_\infty$ | APPROVED | | FS | | JTLA | |
| | AUROC↑ | FPR↓$_{90\%}$ | AUROC↑ | FPR↓$_{90\%}$ | AUROC↑ | FPR↓$_{90\%}$ |
| PGD$^\infty$ | | | | | | |
| $\varepsilon = 0.03125$ | **95.7** | **7.7** | 85.9 | 37.0 | 60.7 | 85.1 |
| $\varepsilon = 0.0625$ | **98.8** | **2.2** | 83.9 | 43.4 | 62.4 | 84.1 |
| $\varepsilon = 0.125$ | **99.5** | **1.2** | 82.4 | 45.7 | 66.2 | 79.0 |
| $\varepsilon = 0.25$ | **99.6** | **0.9** | 81.8 | 45.6 | 67.0 | 78.9 |
| $\varepsilon = 0.5$ | **99.6** | **0.9** | 81.7 | 46.0 | 68.3 | 77.4 |
| BIM | | | | | | |
| $\varepsilon = 0.03125$ | **85.5** | **37.6** | 78.2 | 48.0 | 62.4 | 84.1 |
| $\varepsilon = 0.0625$ | **95.8** | **7.2** | 78.2 | 51.4 | 62.7 | 82.5 |
| $\varepsilon = 0.125$ | **98.6** | **2.4** | 79.0 | 51.1 | 61.1 | 86.3 |
| $\varepsilon = 0.25$ | **99.5** | **1.1** | 80.8 | 47.5 | 65.2 | 78.6 |
| $\varepsilon = 0.5$ | 99.8 | 0.5 | 83.3 | 42.1 | 67.2 | 77.3 |
| FGSM | | | | | | |
| $\varepsilon = 0.03125$ | 75.5 | 74.1 | 64.2 | **62.6** | **78.8** | 69.1 |
| $\varepsilon = 0.0625$ | **82.9** | **53.6** | 59.9 | 65.3 | 82.2 | 62.3 |
| $\varepsilon = 0.125$ | **90.1** | **29.9** | 57.2 | 65.8 | 87.9 | 39.3 |
| $\varepsilon = 0.25$ | **95.8** | **10.0** | 48.6 | 67.6 | 93.8 | 16.7 |
| $\varepsilon = 0.5$ | **99.9** | **0.2** | 54.7 | 64.9 | 98.7 | 2.7 |
| SA | | | | | | |
| $\varepsilon = 0.125$ | **97.6** | **4.0** | 70.3 | 50.3 | 96.0 | 6.7 |
| CW$^\infty$ | | | | | | |
| $\varepsilon = 0.3125$ | **88.4** | **38.1** | 67.8 | 53.1 | 87.3 | 43.6 |
| No Norm | APPROVED | | FS | | JTLA | |
| | AUROC↑ | FPR↓$_{90\%}$ | AUROC↑ | FPR↓$_{90\%}$ | AUROC↑ | FPR↓$_{90\%}$ |
| STA | | | | | | |
| No $\varepsilon$ | **94.3** | **12.6** | 69.5 | 48.4 | 90.3 | 34.8 |

Table 12: AUROC↑ and FPR↓$_{90\%}$ for each considered attack mechanisms, L$_p$-norm constraint and $\varepsilon$ on CIFAR100 for `APPROVED`, FS and JTLA on a ViT-B. The best result for each attack is shown in **bold**.

| CIFAR100 - ViT-B | | | | | | |
|---|---|---|---|---|---|---|
| Norm L1 | APPROVED | | FS | | JTLA | |
| | AUROC↑ | FPR↓$_{90\%}$ | AUROC↑ | FPR↓$_{90\%}$ | AUROC↑ | FPR↓$_{90\%}$ |
| PGD$^1$ | | | | | | |
| $\varepsilon = 50$ | **83.5** | **39.3** | 65.5 | 56.2 | 82.1 | 46.2 |
| $\varepsilon = 60$ | **82.4** | **41.0** | 66.6 | 56.2 | 79.5 | 53.9 |
| $\varepsilon = 70$ | **81.2** | **45.3** | 67.4 | 50.0 | 78.1 | 55.1 |
| $\varepsilon = 80$ | **79.8** | **47.8** | 68.3 | 50.0 | 76.5 | 58.8 |
| $\varepsilon = 90$ | **78.4** | **50.0** | 69.2 | **50.0** | 74.5 | 64.2 |
| $\varepsilon = 100$ | **77.0** | 54.0 | 70.1 | **50.0** | 72.9 | 66.3 |
| $\varepsilon = 500$ | 58.1 | 75.5 | **79.3** | 50.0 | 51.9 | 85.2 |
| $\varepsilon = 1000$ | 78.3 | **44.9** | **80.0** | 62.5 | 52.6 | 86.7 |
| $\varepsilon = 5000$ | **86.1** | **29.4** | 74.0 | 75.0 | 52.4 | 86.2 |
| Norm L2 | APPROVED | | FS | | JTLA | |
| | AUROC↑ | FPR↓$_{90\%}$ | AUROC↑ | FPR↓$_{90\%}$ | AUROC↑ | FPR↓$_{90\%}$ |
| PGD$^2$ | | | | | | |
| $\varepsilon = 0.125$ | 84.3 | 38.3 | 64.6 | 56.2 | **85.5** | **36.0** |
| $\varepsilon = 0.25$ | **82.7** | **41.4** | 66.2 | 56.2 | 80.3 | 50.1 |
| $\varepsilon = 0.5$ | **73.9** | 59.1 | 72.0 | **50.0** | 70.2 | 71.5 |
| $\varepsilon = 5$ | **78.6** | **43.5** | 75.1 | 75.0 | 51.4 | 86.1 |
| $\varepsilon = 10$ | **79.4** | **41.0** | 74.4 | 75.0 | 52.0 | 84.8 |
| HOP | | | | | | |
| $\varepsilon = 0.1$ | **89.1** | **24.8** | 62.7 | 50.0 | 81.7 | 52.1 |
| DeepFool | | | | | | |
| No $\varepsilon$ | **75.5** | 59.9 | 62.2 | **50.0** | 70.5 | 75.1 |
| Norm L$_\infty$ | APPROVED | | FS | | JTLA | |
| | AUROC↑ | FPR↓$_{90\%}$ | AUROC↑ | FPR↓$_{90\%}$ | AUROC↑ | FPR↓$_{90\%}$ |
| PGD$^\infty$ | | | | | | |
| $\varepsilon = 0.03125$ | 75.4 | **51.5** | **76.0** | 74.8 | 48.8 | 87.9 |
| $\varepsilon = 0.0625$ | **88.1** | **26.0** | 68.9 | 75.0 | 53.3 | 83.2 |
| $\varepsilon = 0.125$ | **93.3** | **14.9** | 65.5 | 75.0 | 52.4 | 85.7 |
| $\varepsilon = 0.25$ | **94.4** | **12.8** | 64.3 | 75.0 | 51.3 | 88.4 |
| $\varepsilon = 0.5$ | **89.7** | **26.4** | 64.2 | 75.0 | 52.4 | 84.6 |
| BIM | | | | | | |
| $\varepsilon = 0.03125$ | 63.1 | **72.9** | **67.6** | 75.0 | 51.0 | 87.1 |
| $\varepsilon = 0.0625$ | **70.5** | **64.8** | 63.0 | 81.1 | 51.4 | 85.1 |
| $\varepsilon = 0.125$ | **87.2** | **28.1** | 62.1 | 82.7 | 51.2 | 86.4 |
| $\varepsilon = 0.25$ | **93.2** | **15.4** | 63.7 | 75.4 | 51.5 | 87.0 |
| $\varepsilon = 0.5$ | **96.5** | **8.3** | 65.3 | 75.0 | 52.7 | 86.9 |
| FGSM | | | | | | |
| $\varepsilon = 0.03125$ | **80.8** | **48.1** | 61.9 | 62.5 | 70.9 | 71.1 |
| $\varepsilon = 0.0625$ | **86.5** | **33.0** | 61.3 | 61.4 | 72.7 | 72.0 |
| $\varepsilon = 0.125$ | **90.4** | **24.0** | 54.8 | 50.0 | 77.0 | 65.2 |
| $\varepsilon = 0.25$ | **95.7** | **10.3** | 49.6 | 50.0 | 83.4 | 47.0 |
| $\varepsilon = 0.5$ | **98.6** | **4.1** | 46.2 | 56.2 | 78.7 | 64.0 |
| SA | | | | | | |
| $\varepsilon = 0.125$ | **89.6** | **26.0** | 63.3 | 50.0 | 87.7 | 30.8 |
| CW$^\infty$ | | | | | | |
| $\varepsilon = 0.3125$ | **81.7** | **42.2** | 67.0 | 50.0 | 79.1 | 60.5 |
| No Norm | APPROVED | | FS | | JTLA | |
| | AUROC↑ | FPR↓$_{90\%}$ | AUROC↑ | FPR↓$_{90\%}$ | AUROC↑ | FPR↓$_{90\%}$ |
| STA | | | | | | |
| No $\varepsilon$ | **87.4** | **32.1** | 65.4 | 50.0 | 80.4 | 64.8 |

Table 13: AUROC↑ and FPR↓$_{90\%}$ for each considered attack mechanisms, $L_p$-norm constraint and $\varepsilon$ on CIFAR100 for APPROVED, FS, and JTLA on a ViT-L. The best result for each attack is shown in **bold**.

| CIFAR100 - ViT-L | | | | | | |
|---|---|---|---|---|---|---|
| **Norm L1** | APPROVED | | FS | | JTLA | |
| | AUROC↑ | FPR↓$_{90\%}$ | AUROC↑ | FPR↓$_{90\%}$ | AUROC↑ | FPR↓$_{90\%}$ |
| PGD$^1$ | | | | | | |
| $\varepsilon = 50$ | **87.0** | **30.1** | 61.6 | 75.5 | 85.0 | 43.5 |
| $\varepsilon = 60$ | **87.4** | **29.8** | 61.7 | 75.3 | 84.9 | 42.6 |
| $\varepsilon = 70$ | **87.4** | **28.7** | 61.9 | 75.4 | 84.5 | 42.8 |
| $\varepsilon = 80$ | **87.3** | **29.7** | 62.4 | 75.5 | 83.7 | 42.9 |
| $\varepsilon = 90$ | **87.1** | **29.5** | 62.3 | 75.2 | 81.7 | 49.5 |
| $\varepsilon = 100$ | **86.8** | **30.1** | 62.8 | 74.8 | 82.2 | 47.8 |
| $\varepsilon = 500$ | 59.6 | 70.4 | **63.8** | **68.8** | 58.4 | 81.2 |
| $\varepsilon = 1000$ | 63.9 | 63.2 | **68.9** | **63.0** | 54.4 | 83.7 |
| $\varepsilon = 5000$ | **80.8** | **35.5** | 71.1 | 62.0 | 55.7 | 81.3 |
| **Norm L2** | APPROVED | | FS | | JTLA | |
| | AUROC↑ | FPR↓$_{90\%}$ | AUROC↑ | FPR↓$_{90\%}$ | AUROC↑ | FPR↓$_{90\%}$ |
| PGD$^2$ | | | | | | |
| $\varepsilon = 0.125$ | **86.2** | **31.3** | 61.2 | 75.7 | 84.4 | 43.6 |
| $\varepsilon = 0.25$ | **87.3** | **30.0** | 62.0 | 75.4 | 84.0 | 46.2 |
| $\varepsilon = 0.5$ | **85.2** | **32.2** | 62.9 | 73.8 | 78.1 | 55.5 |
| $\varepsilon = 5$ | **67.9** | **59.3** | 67.6 | 73.8 | 55.5 | 82.9 |
| $\varepsilon = 10$ | **71.0** | **55.2** | 67.5 | 66.4 | 53.8 | 85.9 |
| HOP | | | | | | |
| $\varepsilon = 0.1$ | **91.2** | **21.0** | 55.0 | 74.5 | 86.4 | 44.7 |
| **Norm L$_\infty$** | APPROVED | | FS | | JTLA | |
| | AUROC↑ | FPR↓$_{90\%}$ | AUROC↑ | FPR↓$_{90\%}$ | AUROC↑ | FPR↓$_{90\%}$ |
| PGD$^\infty$ | | | | | | |
| $\varepsilon = 0.03125$ | **71.3** | **54.8** | 70.6 | 61.5 | 54.0 | 83.7 |
| $\varepsilon = 0.0625$ | **87.2** | **25.2** | 71.2 | 61.7 | 55.4 | 80.6 |
| $\varepsilon = 0.125$ | **93.3** | **14.0** | 72.0 | 61.9 | 59.7 | 75.7 |
| $\varepsilon = 0.25$ | **92.1** | **17.9** | 71.7 | 61.2 | 60.2 | 75.8 |
| $\varepsilon = 0.5$ | **95.6** | **10.6** | 71.5 | 61.3 | 59.5 | 77.3 |
| BIM | | | | | | |
| $\varepsilon = 0.03125$ | 59.5 | 82.5 | **64.0** | **69.7** | 52.9 | 85.2 |
| $\varepsilon = 0.0625$ | **75.2** | **48.3** | 65.7 | 69.4 | 55.3 | 81.7 |
| $\varepsilon = 0.125$ | **87.1** | **25.5** | 67.8 | 67.7 | 55.8 | 81.7 |
| $\varepsilon = 0.25$ | **93.5** | **13.6** | 70.5 | 63.7 | 59.5 | 77.3 |
| $\varepsilon = 0.5$ | **97.0** | **7.0** | 73.7 | 57.2 | 61.7 | 72.4 |
| FGSM | | | | | | |
| $\varepsilon = 0.03125$ | **76.2** | **60.5** | 56.1 | 75.5 | 72.0 | 66.3 |
| $\varepsilon = 0.0625$ | **87.2** | **25.2** | 52.8 | 74.3 | 75.7 | 59.6 |
| $\varepsilon = 0.125$ | **89.0** | **28.5** | 46.7 | 73.6 | 81.3 | 52.6 |
| $\varepsilon = 0.25$ | **95.2** | **11.0** | 35.0 | 77.5 | 86.9 | 40.7 |
| $\varepsilon = 0.5$ | **87.5** | **19.6** | 30.6 | 80.6 | 84.6 | 26.7 |
| SA | | | | | | |
| $\varepsilon = 0.125$ | **91.2** | **21.0** | 56.3 | 73.7 | 86.0 | 42.6 |
| CW$^\infty$ | | | | | | |
| $\varepsilon = 0.3125$ | **77.0** | **53.9** | 55.8 | 76.4 | 74.5 | 64.2 |
| **No Norm** | APPROVED | | FS | | JTLA | |
| | AUROC↑ | FPR↓$_{90\%}$ | AUROC↑ | FPR↓$_{90\%}$ | AUROC↑ | FPR↓$_{90\%}$ |
| STA | | | | | | |
| No $\varepsilon$ | **89.4** | **26.5** | 59.2 | 71.1 | 83.3 | 45.5 |

Table 14: AUROC↑ and FPR↓$_{90\%}$ for each considered attack mechanisms, L$_p$-norm constraint and $\varepsilon$ on Tiny ImageNet for APPROVED, FS and JTLA on a ViT-B. The best result for each attack is shown in **bold**.

| | \multicolumn{6}{c}{Tiny ImageNet - ViT-B} | | | | | |
|---|---|---|---|---|---|---|
| Norm L1 | APPROVED | | FS | | JTLA | |
| | AUROC↑ | FPR↓$_{90\%}$ | AUROC↑ | FPR↓$_{90\%}$ | AUROC↑ | FPR↓$_{90\%}$ |
| PGD$^1$ | | | | | | |
| $\varepsilon = 50$ | **74.2** | **61.1** | 44.8 | 81.6 | 66.9 | 69.0 |
| $\varepsilon = 60$ | **74.3** | **60.7** | 45.0 | 81.8 | 69.3 | 64.1 |
| $\varepsilon = 70$ | **74.8** | **60.7** | 45.1 | 82.0 | 68.3 | 65.3 |
| $\varepsilon = 80$ | **74.7** | **60.5** | 45.1 | 82.3 | 70.9 | 66.4 |
| $\varepsilon = 90$ | **74.9** | **59.8** | 45.0 | 82.2 | 70.1 | 63.9 |
| $\varepsilon = 100$ | **74.6** | **59.4** | 44.9 | 82.0 | 69.3 | 66.6 |
| $\varepsilon = 500$ | **76.5** | **59.7** | 60.7 | 71.7 | 70.1 | 66.7 |
| $\varepsilon = 1000$ | **74.2** | **59.4** | 73.7 | 62.4 | 68.5 | 70.0 |
| $\varepsilon = 5000$ | 78.2 | 51.8 | **83.2** | **50.0** | 66.0 | 74.3 |
| Norm L2 | APPROVED | | FS | | JTLA | |
| | AUROC↑ | FPR↓$_{90\%}$ | AUROC↑ | FPR↓$_{90\%}$ | AUROC↑ | FPR↓$_{90\%}$ |
| PGD$^2$ | | | | | | |
| $\varepsilon = 0.125$ | **74.2** | **60.2** | 45.2 | 81.4 | 69.4 | 68.4 |
| $\varepsilon = 0.25$ | **75.0** | **57.2** | 45.2 | 81.8 | 69.3 | 65.6 |
| $\varepsilon = 0.5$ | **75.7** | **53.4** | 47.1 | 79.5 | 70.2 | 64.7 |
| $\varepsilon = 5$ | 74.3 | 60.6 | **77.9** | **57.5** | 66.5 | 73.2 |
| $\varepsilon = 10$ | 74.4 | 59.7 | **78.1** | **57.7** | 65.4 | 73.3 |
| HOP | | | | | | |
| $\varepsilon = 0.1$ | **87.1** | **31.8** | 59.1 | 76.3 | 73.4 | 60.6 |
| Norm L$_\infty$ | APPROVED | | FS | | JTLA | |
| | AUROC | FPR | AUROC | FPR | AUROC | FPR |
| PGD$^\infty$ | | | | | | |
| $\varepsilon = 0.03125$ | 89.6 | 28.8 | **96.0** | **8.2** | 58.9 | 81.5 |
| $\varepsilon = 0.0625$ | **99.1** | **1.9** | 93.8 | 11.9 | 58.8 | 81.5 |
| $\varepsilon = 0.125$ | **99.9** | **0.0** | 89.2 | 47.1 | 60.7 | 76.1 |
| $\varepsilon = 0.25$ | **99.9** | **0.0** | 85.5 | 73.6 | 62.0 | 77.2 |
| $\varepsilon = 0.5$ | **99.9** | **0.0** | 83.6 | 82.2 | 62.0 | 78.8 |
| BIM | | | | | | |
| $\varepsilon = 0.03125$ | 80.7 | **43.1** | **86.0** | 44.8 | 61.2 | 80.4 |
| $\varepsilon = 0.0625$ | **95.1** | **15.1** | 90.3 | 33.4 | 59.1 | 83.3 |
| $\varepsilon = 0.125$ | **99.6** | **1.0** | 87.4 | 61.4 | 58.6 | 80.7 |
| $\varepsilon = 0.25$ | **99.9** | **0.0** | 84.9 | 79.9 | 60.2 | 79.7 |
| $\varepsilon = 0.5$ | **99.9** | **0.0** | 83.9 | 82.5 | 65.5 | 72.6 |
| FGSM | | | | | | |
| $\varepsilon = 0.03125$ | **74.5** | **55.9** | 56.3 | 75.5 | 70.2 | 66.5 |
| $\varepsilon = 0.0625$ | **80.8** | **43.5** | 58.0 | 71.8 | 72.3 | 60.0 |
| $\varepsilon = 0.125$ | **87.1** | **30.4** | 53.6 | 75.1 | 72.7 | 62.1 |
| $\varepsilon = 0.25$ | **91.1** | **22.3** | 48.1 | 78.8 | 74.4 | 59.7 |
| $\varepsilon = 0.5$ | **94.4** | **15.2** | 50.9 | 74.2 | 75.8 | 56.4 |
| SA | | | | | | |
| $\varepsilon = 0.125$ | **77.0** | **49.1** | 48.7 | 78.5 | 70.6 | 63.0 |
| No Norm | APPROVED | | FS | | JTLA | |
| | AUROC | FPR | AUROC | FPR | AUROC | FPR |
| STA | | | | | | |
| No $\varepsilon$ | **80.2** | **42.5** | 53.0 | 77.5 | 68.2 | 68.3 |

Table 15: AUROC↑ and FPR↓$_{90\%}$ for each considered attack mechanisms, L$_p$-norm constraint and $\varepsilon$ on Tiny ImageNet for `APPROVED`, FS, and JTLA on a ViT-L. The best result for each attack is shown in **bold**.

| | | | | | | |
|---|---|---|---|---|---|---|
| Tiny - ViT-L | | | | | | |
| Norm L1 | APPROVED | | FS | | JTLA | |
| | AUROC↑ | FPR↓$_{90\%}$ | AUROC↑ | FPR↓$_{90\%}$ | AUROC↑ | FPR↓$_{90\%}$ |
| PGD[1] | | | | | | |
| $\varepsilon = 50$ | **79.8** | **49.4** | 57.8 | 79.6 | 72.8 | 54.8 |
| $\varepsilon = 60$ | **80.1** | **49.3** | 57.7 | 79.0 | 75.2 | 54.2 |
| $\varepsilon = 70$ | **80.1** | **49.6** | 57.6 | 79.2 | 74.6 | 54.7 |
| $\varepsilon = 80$ | **80.5** | **50.4** | 57.6 | 78.7 | 72.7 | 59.0 |
| $\varepsilon = 90$ | **80.5** | **50.4** | 58.4 | 79.0 | 73.3 | 62.0 |
| $\varepsilon = 100$ | **80.6** | **49.9** | 58.1 | 78.5 | 74.1 | 58.6 |
| $\varepsilon = 500$ | **74.8** | **53.6** | 61.5 | 74.4 | 64.8 | 72.0 |
| $\varepsilon = 1000$ | 65.3 | **67.8** | **66.1** | 68.0 | 58.3 | 79.6 |
| $\varepsilon = 5000$ | 65.8 | 66.3 | **71.2** | **62.2** | 54.6 | 83.4 |
| Norm L2 | APPROVED | | FS | | JTLA | |
| | AUROC↑ | FPR↓$_{90\%}$ | AUROC↑ | FPR↓$_{90\%}$ | AUROC↑ | FPR↓$_{90\%}$ |
| PGD[2] | | | | | | |
| $\varepsilon = 0.125$ | **79.5** | **49.3** | 57.8 | 79.3 | 71.6 | 68.3 |
| $\varepsilon = 0.25$ | **80.1** | **49.5** | 57.7 | 78.7 | 74.1 | 62.0 |
| $\varepsilon = 0.5$ | **80.8** | **48.8** | 57.7 | 79.2 | 73.8 | 58.8 |
| $\varepsilon = 5$ | 64.8 | 71.3 | **65.4** | **69.9** | 57.0 | 82.2 |
| $\varepsilon = 10$ | 64.4 | 71.6 | **65.9** | **69.9** | 58.8 | 81.7 |
| HOP | | | | | | |
| $\varepsilon = 0.1$ | **85.3** | **32.3** | 52.6 | 81.7 | 73.0 | 54.4 |
| DeepFool | | | | | | |
| No $\varepsilon$ | **80.4** | **47.4** | 57.6 | 80.0 | 72.5 | 61.1 |
| Norm L$_\infty$ | APPROVED | | FS | | JTLA | |
| | AUROC↑ | FPR↓$_{90\%}$ | AUROC↑ | FPR↓$_{90\%}$ | AUROC↑ | FPR↓$_{90\%}$ |
| PGD$^\infty$ | | | | | | |
| $\varepsilon = 0.03125$ | 60.4 | 73.6 | **73.3** | **59.5** | 53.8 | 83.6 |
| $\varepsilon = 0.0625$ | 68.8 | 60.5 | **79.6** | **48.9** | 55.4 | 83.4 |
| $\varepsilon = 0.125$ | 79.4 | 40.1 | **82.9** | **43.3** | 59.6 | 81.4 |
| $\varepsilon = 0.25$ | **84.2** | **31.3** | 83.9 | 39.4 | 62.8 | 77.0 |
| $\varepsilon = 0.5$ | **85.7** | **28.6** | 84.3 | 38.8 | 63.5 | 77.6 |
| BIM | | | | | | |
| $\varepsilon = 0.03125$ | 63.8 | 72.2 | **68.1** | **64.0** | 57.2 | 81.7 |
| $\varepsilon = 0.0625$ | 64.6 | 69.3 | **73.9** | **57.1** | 54.3 | 83.6 |
| $\varepsilon = 0.125$ | 72.9 | 53.7 | **79.0** | **49.9** | 57.3 | 80.6 |
| $\varepsilon = 0.25$ | 82.3 | **35.1** | **82.8** | 42.8 | 61.3 | 78.1 |
| $\varepsilon = 0.5$ | **89.8** | **21.4** | 86.9 | 33.3 | 68.8 | 68.2 |
| FGSM | | | | | | |
| $\varepsilon = 0.03125$ | **78.0** | **50.3** | 56.2 | 79.9 | 68.7 | 68.7 |
| $\varepsilon = 0.0625$ | **78.3** | **48.5** | 55.4 | 79.7 | 69.9 | 66.0 |
| $\varepsilon = 0.125$ | **79.5** | **48.5** | 53.7 | 81.0 | 69.8 | 64.2 |
| $\varepsilon = 0.25$ | **82.6** | **38.2** | 53.1 | 80.0 | 71.8 | 57.7 |
| $\varepsilon = 0.5$ | **78.8** | 44.1 | 50.9 | 75.4 | 75.2 | **46.9** |
| SA | | | | | | |
| $\varepsilon = 0.125$ | **82.3** | **32.3** | 56.3 | 82.1 | 75.0 | 50.6 |
| CW$^\infty$ | | | | | | |
| $\varepsilon = 0.3125$ | **73.7** | **57.6** | 51.9 | 83.9 | 66.7 | 70.4 |
| No Norm | APPROVED | | FS | | JTLA | |
| | AUROC↑ | FPR↓$_{90\%}$ | AUROC↑ | FPR↓$_{90\%}$ | AUROC↑ | FPR↓$_{90\%}$ |
| STA | | | | | | |
| No $\varepsilon$ | **85.7** | **30.7** | 63.4 | 69.3 | 74.1 | 52.4 |

