# OpenReview forum: "A Simple Unsupervised Data Depth-based Method to Detect Adversarial Images"
_TMLR — Rejected by TMLR_

### Review · Reviewer_vaAY · 2023-04-11

**Summary Of Contributions:**

The paper presents an unsupervised method called APPROVED to detect adversarial images in vision transformers for image classification. The method leverages data depth to compute the similarity between an input sample and the training distribution, making it efficient and computationally inexpensive. In contrast to existing methods, which rely on additional training or specific heuristics, APPROVED is simple and has a geometrical interpretation. The paper contributes to the field by introducing the use of data depths for detecting adversarial images in transformers, and by providing an experimental setting that goes beyond ResNet architectures. The authors also provide their code. APPROVED shows promising results on CIFAR10, CIFAR100, and Tiny ImageNet datasets.

**Audience:**

Yes

**Broader Impact Concerns:**

The authors provided some limitations as well as a broader impact.

**Claims And Evidence:**

Yes

**Requested Changes:**

I would like the authors to address the points listed in my weakness section above.

**Strengths And Weaknesses:**

## Strengths
(+) The detection of adversarial examples is an important research topic toward the robustness and generalization of adversarial examples.
(+) To the best of my knowledge, this is the first work that leverages “data depth” to detect adversarial examples.
(+) This is one of a few works which investigate the detection of adversarial examples on vision transformers.
(+) This work is clearly written, easy to follow and technically sound to the best of my judgment.

## Weaknesses
(-) Previous works have demonstrated that most adversarial detection techniques can be fooled [A, Athalye et al, B, C], e.g. due to obfuscated gradients. While the authors dedicate Section 5.2 “Adaptive Attacks” to that topic, unfortunately, they brush off the most important part: the evaluation of white-box adaptive attacks. The authors justify this with the statement that “finding a differentiable surrogate of IRW remains a very challenging open research question”. At least a more fine-grained discussion is required to justify this claim. The authors should also demonstrate that the approximation of the non-differentiable part through a neural network (or other techniques) indeed fails.
(-) [Athalye et al, B] provide guidelines for the evaluation of new adversarial defense techniques. The authors should follow & discuss these guidelines.
(-) I noticed a possible flaw in the experimental setup. The authors stated, “Indeed, the input of a ViT has more pixels than the input of a ResNet (32 × 32 × 3 for ResNet and 224 × 224 × 3 for ViT).” This suggests that the authors used input sizes `32x32x3` and `224x224x3` for ResNet and ViT, respectively for CIFAR. I guess the authors upsampled the images to `224x224x3` for ViT for CIFAR. This seems not like a fair comparison. This might be a simple misunderstanding on my side but I would like to ask the authors to elaborate on that. I further suggest using the same input size for both networks, either `32x32x3` or `224x224x3` for CIFAR in their experimental setup.
(-) It seems that the proposed technique is only applicable to ViTs. I am wondering if this phenomenon is only observed for this specific ViT architecture. Since the authors only evaluate ViT it is not clear if this is a general observation for transformer architectures. Additionally, I am wondering about the data depth values of other standard architectures, such as DenseNet, Inception, MobileNet, or MLP-Mixer.
(-) Regarding white-box attacks, I would like to note that AutoAttack is the current best practice to evaluate white-box attacks.
(Minor) In Figure 2 “top”, and “bottom”, should be changed to “left”, and “right”.

[A] Adversarial Examples Are Not Easily Detected: Bypassing Ten Detection Methods; AISec’17
[B] On Adaptive Attacks to Adversarial Example Defenses; NeurIPS 2020
[C] Detecting Adversarial Examples Is (Nearly) As Hard As Classifying Them; ICML 2021 workshop

---

> ### Author Response · Authors · 2023-05-03
> **Reply vaAY**
>
>
> First, let us thank vaAY for their careful reading of our manuscript. Below we address the reviewer’s concerns.
>
> **Adaptive attacks**. We kindly refer the reviewer to the second point of the general comment.
>
> **Image Dimension**. We kindly refer the reviewer to the first point of the general comment.
>
>
> **Other models**. Reviewer vaAY suggests that our method could be extended to other types of classifiers. While we did not explicitly claim generalization to different architectures in our paper, we agree with the reviewer that our method has the potential to be applied to other models. Our focus on ViT was motivated by the trend in the community towards the use of transformers, as also acknowledged by reviewer NrUG and highlighted in recent works such as Dehghani et al. (2023). Nevertheless, we acknowledge that the applicability of our method to other architectures is an interesting avenue for future research, and **we plan to investigate this further**.
>
> **On the use of AutoAttack**.  Regarding the suggestion by reviewer vaAY to use AutoAttack to evaluate our methods, we would like to respectfully disagree. While AutoAttack is a state-of-the-art benchmark for testing robust classifiers, we believe it is not the most appropriate fit for benchmarking adversarial attack detectors. It is worth noting that AutoAttack is composed of four different attacks, two of which are slightly modified versions of the PGD algorithm that we already consider, another is a slightly modified version of DF, and the final one is Square Attack. Furthermore, AutoAttack is based on the worst-case scenario, meaning that if one attack is successful for a specific sample, the others will not be considered. Since our underlying classifier is defenseless, PGD is already strong enough to attack almost every sample. Therefore, AutoAttack will end up being just A-PGD. Thus, evaluating each attack separately, as we did, will provide a better sense of security than considering AutoAttack directly. **We added this explanation in the Appendix along with the relevant citations to the related work.**
>
> We hope we have addressed reviewer vaAY’s concerns. We are happy to further discuss any remaining questions.

---

### Review · Reviewer_NrUG · 2023-04-15

**Summary Of Contributions:**

This paper explores the possibility of using data depths, a statistical tool with geometrical intrepretation, as a metric to detect adversarial example. The paper performs layerwise analysis and compares ViT and CNN models, and find the proposed data depths method best suitable to be applied on the logits layer of ViT models. The proposed method shows improved detection performance than FS and JTLA on multiple attacks performed on ViT models.

**Audience:**

Yes

**Broader Impact Concerns:**

No concern on broader impact

**Claims And Evidence:**

No

**Requested Changes:**

1. Address point 1 and 2 in the weakness by providing more information in the intro and related work sections
2. Evaluate the proposed method with stronger attack like AutoAttack, and the potential adaptive attack mentioned in weakness point 3
3. Fix the attack strength comparison by conducting both ViT and ResNet attacks on the 32x32 input space
4. Fix minor issues: (1) Typo frac1n in Equ. (6); (2) Explain which Lp each subfigure corresponds to in Fig. 1; (3) Instead of plotting difference, it would be more interesting to directly plot the adversarial and clean image depth distribution to show overlap in Fig. 2; (4) In Tab 5 and 6 JLTA results are provided, but table captions say MagNet.

**Strengths And Weaknesses:**

## Strength
1. This paper introduces the metric of data depth into adversarial attack detection, which is a novel metric for this field
2. The focus on ViT model, with discussion on the difference between ViT and ResNet models should be encouraged as SOTA classification models tends to take ViT architecture.

## Weakness
1. The use of data depth is not particularly designed based on the characteristic of adversarial attack. More intuiation shoudl be provided on why adversarial attack may lead to smaller data depth, especially in ViT logits layers.
2. The discussion of related work is inadequate. For Sec. 2.1, recent strong attack like AutoAttack [1] should be discussed and utilized in the experiment. For Sec. 2.2, the limitation of detection methods should also be discussed, especially facing adaptive attacks. Notable previous work like "bypassing 10 detection methods" [2] should be mentioned
3. The author claims finding differentiable surrogate of IRW challenging. However, with the knowledge of the defense mechanism simple attacks can be designed to challenge the proposed method. Since IRW directly considers similarity of a point to a distribution, the most straightforward surrogate would be mapping the attack logits to the clean logits distribution. For example, previous work [3] shows it effective to generate attack by matching the attack feature to the clean feature of a target image in a hidden layer of the model. Therefore an adaptive attack of the proposed method can be conducted as matching adversarial logits with a targeted training example logits, therefore achieving an in-distribution logits for the attack. The attack objective can be formulated as $x'=\arg\min_{||x'-x||_p<\epsilon} ||f^{L-1}(x') - f^{L-1}(\hat{x})||_2^2$, where $\hat{x}$ is a target image in the training set with label $\hat{y}$.
4. Though may not relevant to the contribution of the paper, the discussion in Sec 4.2 is not fair and misleading. A fair comparison between ResNet and ViT attack strength should be conducted at the same input dimension. For ViT the attack should be added to the 32x32 input before the input is scaled to 224x224 dimension. This will give more representative strength for future work to analysis on ViT attacks
5. There are minor typos and formatting errors, see requested changes

### References
[1] Croce, F., & Hein, M. (2020, November). Reliable evaluation of adversarial robustness with an ensemble of diverse parameter-free attacks. In International conference on machine learning (pp. 2206-2216). PMLR.

[2] Carlini, N., & Wagner, D. (2017, November). Adversarial examples are not easily detected: Bypassing ten detection methods. In Proceedings of the 10th ACM workshop on artificial intelligence and security (pp. 3-14).

[3] Inkawhich, N., Wen, W., Li, H. H., & Chen, Y. (2019). Feature space perturbations yield more transferable adversarial examples. In Proceedings of the IEEE/CVF Conference on Computer Vision and Pattern Recognition (pp. 7066-7074).

---

> ### Author Response · Authors · 2023-05-03
> **Reply to NrUG**
>
> First, let us thank NrUG for their careful reading of our manuscript. Below we address the reviewer’s concerns.
>
> **On the intuition of data–depth & updated related work**.
>
> To provide a better sense of why data depths are useful tools to distinguish between natural and attacked samples, we decided to provide a graphical explanation. See Appendix C, ‘‘Intuition on data depths’’.
>
> On CIFAR-10, using a ViT-B_16 as the underlying classifier, we plotted the training, natural and adversarial logits for class 0 and class 1 output by the network, using a t-SNE to reduce the dimension to 2. The visualization is provided in Figure 6 in the Appendix.
>
> From this figure, we observe that the natural testing samples are deep inside the training distribution while the adversarial samples are on the border. It shows that data depth, which computes the depth of an element w.r.t. a reference distribution, fit to distinguish between clean and attacked samples and is well adapted to this problem.
>
> **This remark is added in the Appendix of our manuscript. Also, we have added a reference to adaptive attacks in Section 2.2.**
>
>
>
> **On the use of AutoAttack**.  Regarding the suggestion by reviewer NrUG to use AutoAttack to evaluate our methods, we would like to respectfully disagree. While AutoAttack is a state-of-the-art benchmark for testing robust classifiers, we believe it is not the most appropriate fit for benchmarking adversarial attack detectors. It is worth noting that AutoAttack is composed of four different attacks, two of which are slightly modified versions of the PGD algorithm that we already consider, another is a slightly modified version of DF, and the final one is Square Attack. Furthermore, AutoAttack is based on the worst-case scenario, meaning that if one attack is successful for a specific sample, the others will not be considered. Since our underlying classifier is defenseless, PGD is already strong enough to attack almost every sample. Therefore, AutoAttack will end up being just A-PGD. Thus, evaluating each attack separately, as we did, will provide a better sense of security than considering AutoAttack directly. **We added this explanation in the Appendix along with the relevant citations to the related work.**
>
>
> **Adaptive attacks**. We kindly refer the reviewer to the second point of the general comment.
>
> **Image Dimension**. We kindly refer the reviewer to the first point of the general comment.
>
> We have fixed the typos.
>
> *We hope we have addressed reviewer NrUG’s concerns. We are happy to further discuss any remaining questions.*

---

### Review · Reviewer_jxr3 · 2023-04-25

**Summary Of Contributions:**

The paper proposes a method, APPROVED, to detect adversarially perturbed images. It relies on data depth to compute a similarity score of the logits of training points and new inputs. It doesn't require to be jointly trained with the classifier, and it is not limited to a specific threat model. Moreover, it is particularly well-suited for vision transformers. In the experiments, APPROVED outperforms existing detection methods on various datasets.

**Audience:**

Yes

**Broader Impact Concerns:**

No concern.

**Claims And Evidence:**

No

**Requested Changes:**

I think the evaluation needs to include stronger adaptive attacks. Moreover, the presentation should be improved, and the missing information included (see above).

**Strengths And Weaknesses:**

Strengths
- Using data depth is, as far as I know, a novel approach for detection.

- Being agnostic of both training and threat model is an appealing property for a detection method.

Weaknesses
- The proposed method measures the similarity of the logits of an input with those of the training points belonging to the predicted class. Then, a simple adaptive attack could try to match the logits of the adversarial points to the logits of some random training points of one of the classes other than the correct one, or the average of the logits of the training points of one class. This could be optimized via PGD or similar methods. Moreover, the paper mentions that having a non-differentiable component should provide more robustness, but previous works have shown that this often only makes an adaptive attack necessary, since standard ones fail (Athalye et al., 2018, Tramèr et al., 2020).

- In general, it is unclear how strong the evaluation setup is: many details about the attacks are missing (number of iterations and restarts, loss functions used, etc.). Also, the attacks in the $\ell_\infty$ threat model are used with $\varepsilon$ up to 0.5, which can completely delete the original image (since it's in $[0, 1]^d$): this violates the definition of adversarial perturbations, which should preserve the true class of the image, and might lead to easier detection. Finally, using different input resolution for ResNets and ViTs seems to make the results not comparable: if ViTs include an up-scaling step, I think it'd be better to include it in the model, so that the attacks are carried out at the original resolution.

- The presentation needs improvements: some details are missing e.g. the value of $n_{\textrm{proj}}$, and the performance on clean points. Also, the tables in the main part have too small font size, which makes them hardly readable, and in the figures axes labels and plots titles are missing.

---

> ### Author Response · Authors · 2023-05-03
> **Reply to jxr3**
>
> First, let us thank jxR3 for their careful reading of our manuscript. Below we address the reviewer’s concerns.
>
> **Adaptive attacks**. We kindly refer the reviewer to the second point of the general comment.
>
> **On the evaluation setup**.
> We have taken two steps to address the reviewers' concerns. Firstly, we have added the missing parameters of the attacks in Appendix A, and we plan to make our code publicly available upon acceptance. Secondly, we recognize that there are multiple definitions of adversarial attacks. While some definitions require attacks to be invisible to the human eye, others, such as DF [1], FAB [2], C&W [3], have no limitations on the maximum allowed perturbation. Furthermore, even for classical datasets, attacks can be perceptible (e.g., the classical $\varepsilon$ value for MNIST is 0.3125 [4-6], which is relatively high). Therefore, in our paper, we have chosen to consider a wide range of $\varepsilon$ values to cover all possible scenarios. **We have added this explanation at the beginning of Section 4 in our paper.**
>
> **Presentation improvements**:
> As suggested by the reviewer, we have included the values of $n_proj$ and other additional hyperparameters in our updated version.
>
> *We hope we have addressed reviewer jxR3’s concerns. We are happy to further discuss any remaining questions.*
>
> **References:**
>
>
> [1] Seyed-Mohsen Moosavi-Dezfooli, Alhussein Fawzi, and Pascal Frossard.Deepfool: a simple and accurate method to fool deep neural networks. In CVPR 2016.
>
> [2] Nicholas Carlini and David Wagner. Towards evaluating the robustness of neural networks. In 2017 IEEE Symposium on Security and Privacy (SP).
>
> [3] Francesco Croce and Matthias Hein. Minimally distorted adversarial examples with a fast adaptive boundary attack. In International Conference on Machine Learning, 2020.
>
> [4] Hongyang Zhang, Yaodong Yu, Jiantao Jiao, Eric P Xing, Laurent El Ghaoui, and Michael I Jordan. Theoretically principled trade-off between robustness and accuracy. In International Conference on Machine Learning, 2019.
>
> [5] Aleksander Madry, Aleksandar Makelov, Ludwig Schmidt, Dimitris Tsipras, and Adrian Vladu. Towards deep learning models resistant to adversarial attacks. In International Conference on Learning Representations, 2018
>
> [6] Ian J Goodfellow, Jonathon Shlens, and Christian Szegedy. Explaining and harnessing adversarial examples. arXiv preprint arXiv:1412.6572, 2014.

---

### Author Response · Authors · 2023-05-03
**General Answer**

**We are pleased that all reviewers expressed interest in our method, particularly its agnostic-to-threat approach (reviewer jxR3) and the novelty of the data-depth tool we introduced (reviewers jxR3, vaAY, and NrUG). We are also glad that the technical soundness of our work was recognized (reviewer vaAY) and that the use of ViT was deemed interesting, given the limited research (reviewer vaAY) on this topic and the growing popularity of transformers in the community (reviewer NrUG).**

Besides the question of extending our work to computer vision, **we are happy to see that reviewers find very few flaws.** The questions raised by the reviewers mainly focus on three aspects:
1. **Image Resolution in Section 4.**
To clarify, the objective of Section 4 is not to compare the attack strength of ViT and ResNet. Instead, the focus of this section is two-fold: (1) to customize the attack strength of different methods by determining the appropriate $\varepsilon$ values when modifying the image resolution and (2) to justify the selection of the last layer in ViT by identifying the relevant information necessary for detecting attacks.
To address the concerns raised by the reviewers, we have taken two steps: (1) added a clarification at the beginning of Section 4.2 concerning the goal of the section and (2) updated Figure 2 to include the information distribution using a ResNet50 with the image resolution of 224x224x3. **Notice that the previous analysis remains valid.**
2. **Adaptive attacks.**
Following the reviewers’ suggestions, we have improved the evaluation under adaptive attacks by using $\arg\min_{\|x-x^{\prime}\|} \| f(x^{\prime}) - f(x_t)\|^2_2$ as the objective of the attacker, as asked by reviewer NrUG. We used the PGD algorithm, as suggested by review jxR3, to craft our attacks.
We present the results of our experiments in Table 9. While it is true that APPROVED experiences a decline in performance under this specific adaptive attack, our method does not collapse. It performs similarly under adaptive attacks as JTLA does under regular attacks. **We believe that this experiment serves to further validate the effectiveness of our method.**
3. **Typos.**
We would like to express our gratitude to the reviewers for their meticulous reading. We have addressed the typos in the updated version of the paper.

---

### Decision · Action_Editors · 2023-06-05

**Recommendation:** Reject

**Comment:**

3 expert reviewers have reviewed this work. After the submitted the rebuttal by the authors, 1 reviewer gave an acceptance rating while 2 reviewers gave a rejection rating. The main focus of the reviewers were the algorithm's performance under adaptive attacks. While the authors showed results with adaptive attacks in the rebuttal, 2 of the reviewers were not satisfied with the quality of the results as well as a detailed discussion of the setup of the experiments and some important technical discussions. Noting that all reviewers have emphasized the importance of the results on adaptive attacks, with the reviewer opinions more concentrated on the side that the current results are not good/comprehensive enough, the AE hence recommends rejection of this paper.

**Audience:**

Yes. The paper covers adversarial attacks to deep networks which is within scope.

**Claims And Evidence:**

The paper proposes a method, APPROVED, to detect adversarially perturbed images. It relies on data depth to compute a similarity score of the logits of training points and new inputs. It doesn't require to be jointly trained with the classifier, and it is not limited to a specific threat model. Moreover, it is particularly well-suited for vision transformers. In the experiments, APPROVED outperforms existing detection methods on various datasets.

As reviewers pointed out, the claim is somewhat weakened by the paper's lack of capability to satisfactorily deal with adaptive attacks.

**Resubmission Of Major Revision:**

The authors may consider submitting a major revision at a later time.